# A prophage-encoded sRNA limits phage infection of adherent-invasive *E. coli*

Nicole L. Pershing[1,2☯], Robert S. Brzozowski[3☯], Amelia K. Schmidt[3☯], Dominick R. Faith[4,5], Alex C. Joyce[4,5], Lizett Ortiz de Ora[4,5], John D. Kominsky[4], Annika Dankwardt[1], Andrew Maciver[3], Rickesha Bell[6], William S. Henriques[4], Shelby E. Andersen[7], Blake Wiedenheft[4], Sherwood R. Casjens[2], Breck A. Duerkop[7], June L. Round[2]*, Patrick R. Secor [4,5]*

1 Department of Pediatrics, Division of Pediatric Infectious Diseases, University of Utah School of Medicine, Salt Lake City, Utah, United States of America, 2 Department of Pathology, Division of Microbiology and Immunology, University of Utah School of Medicine, Huntsman Cancer Institute, Salt Lake City, Utah, United States of America, 3 Division of Biological Sciences, University of Montana, Missoula, Montana, United States of America, 4 Department of Microbiology & Cell Biology, Montana State University, Bozeman, Montana, United States of America, 5 Center for Biofilm Engineering, Montana State University, Bozeman, Montana, United States of America, 6 Office of Comparative Medicine, University of Utah, Salt Lake City, Utah, United States of America, 7 Department of Immunology and Microbiology, School of Medicine, University of Colorado – Anschutz Medical Campus, Aurora, Colorado, United States of America

☯ These authors contributed equally to this work.
* june.round@path.utah.edu (JLR); patrick.secor@montana.edu (PRS)

## Abstract

Prophages are prevalent features of bacterial genomes that can reduce susceptibility to infection by competing phages, yet the mechanisms involved are often elusive. Here, we identify a small RNA (svsR) encoded by the lambdoid prophage NC-SV in adherent-invasive *Escherichia coli* strain NC101 that limits infection by virulent coliphages. Comparative genomics revealed that NC-SV–like prophages and *svsR* homologs are broadly conserved across Enterobacteriaceae. Transcriptomic analyses show that svsR represses maltodextrin transport genes, including *lamB*, which encodes the outer membrane maltoporin LamB, a known receptor for numerous coliphages. Deletion of the *lamB* gene reveals that while LamB is not required for replication of the virulent phages tested, it contributes to plaque expansion, indicating a role in phage spread but not as an essential receptor. Nutrient supplementation experiments further linked maltodextrin and glucose availability to changes in plaque expansion and phage adsorption. *In vivo,* we compared wild-type NC101 and a prophage-deletion strain (NC101$^{\Delta NC\text{-}SV}$) in mice to assess the impact of NC-SV on lytic phage susceptibility. Although intestinal *E. coli* densities remained stable across groups, animals colonized with NC101 exhibited markedly reduced phage burdens in both the intestinal lumen and mucosa compared to mice colonized with NC101$^{\Delta NC\text{-}SV}$. This reduced phage pressure was associated with increased dissemination of *E. coli* to extraintestinal tissues, including the spleen and liver. Together, these findings highlight a nutrient-responsive,

**Data availability statement:** Raw sequencing reads have been deposited as part of BioProject PRJNA1225266 in the NCBI SRA database.

**Funding:** This work was supported by NIH grant R01DK124317 to J.L.R and P.R.S. P.R.S. was also supported by NIH grant P20GM103474. D.R.F. was supported by the NSF GRFP (366502). N.L.P was supported by the National Center for Advancing Translational Sciences of the National Institutes of Health under Award Numbers UM1TR004409 and K12TR004413 and a pilot grant from the Intermountain Foundation at Primary Children's Hospital. The content is solely the responsibility of the authors and does not necessarily represent the official views of the National Institutes of Health or other funding sources. Research in the Wiedenheft laboratory is supported by the National Institutes of Health (NIH R35GM134867), the M.J. Murdock Charitable Trust and the Montana Agricultural Experimental Station. Funding for the Montana State University Cryo-EM Core Facility (RRID: SCR_026324) was contributed by the National Science Foundation (DBI-1828765), the M.J. Murdock Charitable Trust, NIGMS (P30GM140963) and the MSU Office of Research, Economic Development and Graduate Education. The funders had no role in study design, data collection and analysis, decision to publish, or preparation of the manuscript.

**Competing interests:** The authors have declared that no competing interests exist.

prophage-encoded mechanism that protects *E. coli* from phage predation and may promote bacterial persistence in and dissemination from the mammalian gut.

## Author summary

Prophages—viral genomes integrated into bacterial chromosomes—are common in enteric bacteria and can profoundly influence bacterial physiology and ecological fitness. Here we show that a lambdoid prophage, NC-SV, in the adherent-invasive Escherichia coli strain NC101 modulates host metabolism to limit infection by virulent coliphages. NC-SV encodes a small regulatory RNA, svsR, that re-presses maltodextrin transport genes, including *lamB*, which encodes a maltoporin frequently used as a phage receptor. Through this transcriptional reprogramming, NC-SV reduces phage adsorption and replication both *in vitro* and in the mouse intestine. Consequently, mice colonized with NC101 exhibited markedly lower phage loads in the gut compared with those colonized by NC101 cured of the NC-SV prophage, and this reduction in intestinal phage burden was associated with greater dissemination of NC101 to extraintestinal sites such as the liver and spleen. These findings reveal a nutrient-responsive mechanism of phage resistance mediated by a prophage-encoded sRNA and illustrate how prophages can rewire bacterial regulatory networks in ways that alter phage–host dynamics in the gut.

## Introduction

Bacteriophages (phages) are abundant viruses that infect bacteria and play a key role in shaping microbial communities. Their impact is especially significant in densely populated environments like the gut. Many phages exist as prophages—viral genomes integrated into bacterial chromosomes—that often provide competitive advantages to their bacterial hosts. One well-established function of prophages is protection against competing phages, enhancing host survival in phage-rich niches [1–9]. However, while several prophage-encoded exclusion systems (such as the λ CI repressor) are well characterized, the diversity and molecular details of many other prophage-mediated resistance mechanisms remain less well understood.

Environmental factors such as diet shape the gut microbiota and contribute to dysbiosis in inflammatory bowel diseases (IBD) like Crohn's disease and ulcerative colitis [10–12]. One example is maltodextrin, a common dietary additive which degrades mucus integrity, exacerbates inflammation [13,14], and promotes the expansion of adherent-invasive *Escherichia coli* (AIEC) [15,16], a pathobiont enriched in the IBD gut [10,11,17,18]. Maltodextrin uptake in *E. coli* is mediated by the outer membrane maltoporin LamB, a known receptor for several coliphages including lambda and others [19–22]. Thus, metabolic responses to dietary components can directly influence bacterial surface physiology in ways that may alter phage susceptibility and bacterial fitness in phage-rich environments.

As enteric bacteria proliferate in the inflamed gut, so too do phages that target them [6,7,23–25]. This expansion of enteric phages can reshape microbial communities and potentially influence IBD progression [24,26,27]. Despite this, how pathobionts like AIEC persist and replicate amid rising phage pressure remains unclear.

In this study, we identify a lambda-like prophage, NC-SV, in AIEC strain NC101 that confers protection against some phages by reducing virion adsorption to the bacterial surface. Our data demonstrate that while the NC-SV prophage is dispensable for bacterial growth *in vitro* and *in vivo* in the absence of virulent phage pressure, it confers resistance to virulent phage replication in both contexts. Using RNA sequencing, we found that NC-SV represses maltodextrin transport genes, including *lamB* [21,22]. Previous studies have shown that maltodextrin upregulates *lamB* expression while glucose represses it via catabolite repression [28]. Consistent with this, we observed that maltodextrin enhances phage adsorption, whereas glucose reduces it. Plaque size analyses reveals that some virulent phages form larger plaques on the NC101$^{\Delta NC-SV}$ strain than on wild-type NC101, and plaque sizes contract when *lamB* is deleted from the NC101$^{\Delta NC-SV}$ background. We further identified a prophage-encoded small RNA (sRNA) that phenocopies the whole NC-SV prophage deletion strain by repressing *lamB* and other maltodextrin transport genes and reducing phage adsorption and plaque size.

Together, our findings reveal a mechanism by which the NC-SV prophage modulates host nutrient uptake and surface receptor gene expression to protect against viral infection. This strategy may be particularly advantageous to Enterobacteriaceae in the dynamic environment of the gut where persistence requires balancing stressors of variable nutrient availability and phage predation [29–36].

## Results

### The NC-SV prophage protects *E. coli* NC101 from phage infection by reducing virion adsorption

Many phages that inhabit the gut reside as prophages integrated into the genome of their bacterial host [37]. The genome of AIEC strain NC101 contains three intact prophages with three distinct morphotypes including Siphovirus (NC-SV), Myovirus (NC-MV), and Inovirus (NC-Ino) (Table 1). NC-SV is related to phage lambda and is integrated at the phage 21 attachment site within the isocitrate dehydrogenase gene (*icd*) in the *E. coli* NC101 chromosome. NC-SV encodes lambda-homologous head, tail, DNA replication, homologous recombination, lysis, integration, and regulatory gene modules, which are arranged in the same genomic order as in lambda, supporting their evolutionary relatedness (Fig A in S1 Text). NC-MV is integrated between *E. coli* genes *ybjL* and *rcdA*. NC-Ino is a previously undescribed filamentous *Inoviridae* phage. Some *E. coli* genomes (*e.g.,* strain SF-468, accession No. CP012625) are neatly missing the 9,335 bp Inovirus prophage while other bacteria (*e.g., Yersinia pestis* CO92 and *Salmonella enterica* AR_0127 as well as *Shigella*, *Citrobacter* and *Proteus* strains) carry very similar *Inoviridae* prophage elements in different genomic contexts.

Prophages frequently encode mechanisms that protect their bacterial host from infection by competing phages [5,8]. We hypothesized that one or more of the intact prophages listed in Table 1 would protect NC101 from phage infection. To test this hypothesis, we deleted each prophage from the NC101 chromosome and challenged with virulent phages isolated from wastewater (phages RSB01–04). Genomic analyses of the RSB wastewater phages reveal that RSB01 and RSB03 are both Veterinaerplatzviruses that share 51.7% nucleotide homology while RSB02 and RSB04 are related Kuraviruses that share 60.7% nucleotide homology (Fig B in S1 Text). Kuraviruses are exemplified by *E. coli* phage φEco32,

**Table 1. Prophages in the *E. coli* NC101 chromosome.**

| Prophage | NC101 (locus tag) range | Prophage size (bp) | Phage morphotype | Comments |
|---|---|---|---|---|
| NC-SV | ECNC101_17984 - _18284 | 46,386 | Siphovirus | Related to dsDNA phage lambda |
| NC-MV | ECNC101_12352 - _12582 | 35,247 | Myovirus | Related to dsDNA phage P2 |
| NC-Ino | ECNC101_16337 - _16297 | 9,335 | Inovirus | Filamentous ssDNA phage |

whose lifecycle has been characterized [38,39], and are phylogenetically related to Veterinaerplatzviruses, which form a distinct but allied clade within the same viral class (Caudoviricetes).

No differences in PFUs were observed under any condition except for a modest increase with RSB04 on NC101$^{\Delta NC-MV}$ (Fig C in S1 Text). However, we observed an increase in plaque size for phages RSB01 and RSB03 on NC101$^{\Delta NC-SV}$ compared to wild-type cells, while no significant differences were observed for RSB02 and RSB04 in any strain tested (Figs 1A, 1B, and D in S1 Text). In liquid culture, RSB01 and RSB03 were also more virulent against NC101$^{\Delta NC-SV}$ than against wild-type NC101 (Fig 1C), whereas no differences in virulence were observed for RSB02 and RSB04 in any strain tested (Fig D in S1 Text). Additionally, we did not observe any growth defects in uninfected cultures of any prophage mutant compared to wild-type NC101 (Figs 1C and D in S1 Text).

An increase in plaque size for phages RSB01 and RSB03 but not RSB02 or RSB04 suggests that a cell surface receptor used by RSB01 and RSB03 may be differentially regulated by NC-SV, which could affect phage adsorption to host cells. To test this, we performed phage adsorption assays on NC101 and NC101$^{\Delta NC-SV}$. Phage RSB01 and RSB03 adsorption was significantly (P < 0.01) higher in NC101$^{\Delta NC-SV}$ compared to NC101 cells (Figs 1D and D in S1 Text) while adsorption of phages RSB02 and RSB04 was not affected (Fig D in S1 Text), suggesting that RSB02 and RSB04 use a different cell surface receptor than RSB01 and RSB03. Examination of RSB03-infected cells by transmission electron microscopy revealed that NC101$^{\Delta NC-SV}$ cells appear more prone to lysis by RSB03 compared to NC101 cells (Fig 1E). Overall, these

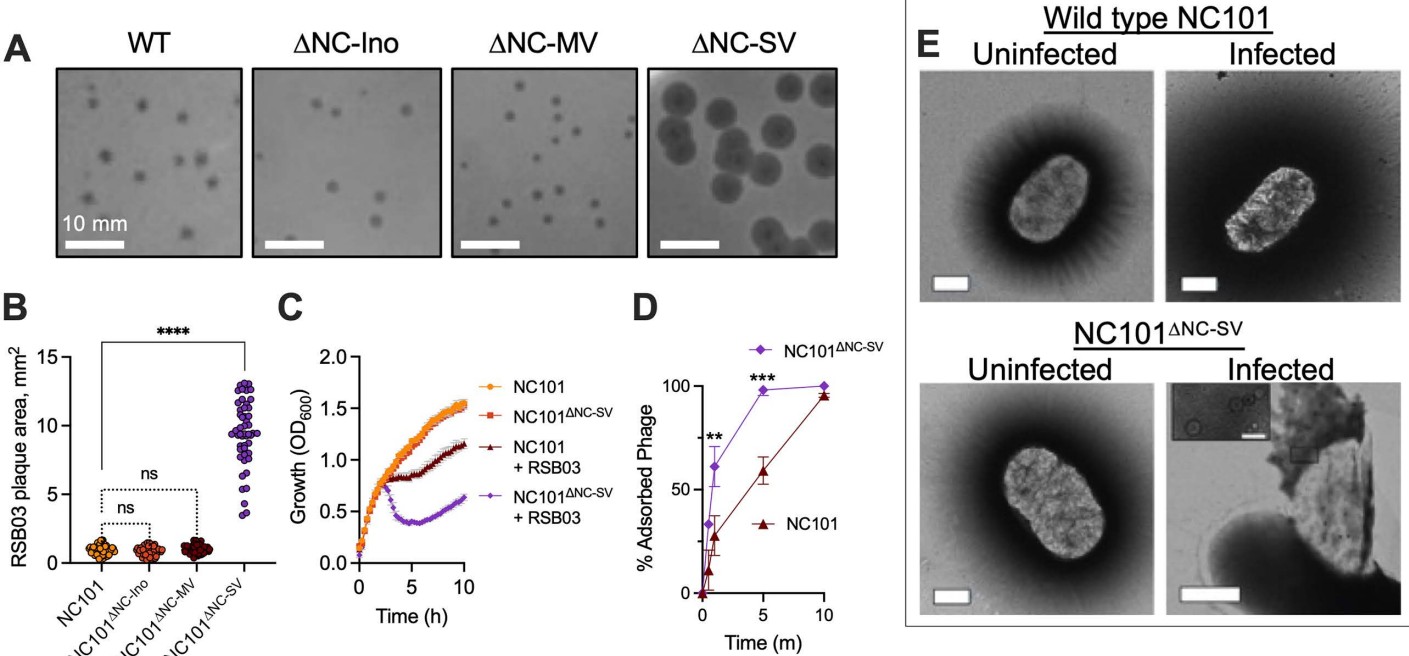

**Fig 1. The NC-SV prophage protects *E. coli* NC101 from phage infection by reducing virion adsorption. (A)** Plaques formed by wastewater phage isolate RSB03 on *E. coli* NC101 or the indicated prophage mutants. Representative images are shown. Scale bars represent 10 mm. **(B)** The surface area of RSB03 plaques formed on lawns of the indicated strains was measured, N = 50 plaques per condition, from four replicate experiments, ****P < 0.0001 compared to NC101; ns, not significant. **(C)** Growth (OD$_{600}$) of NC101 and NC101$^{\Delta NC-SV}$ was measured after infection with RSB03 (MOI = 1). Data are the mean of four experiments. **(D)** The percentage of adsorbed RSB03 virions was measured in NC101 or NC101$^{\Delta NC-SV}$ cells at the indicated times post infection, MOI = 1. Data are the mean ± SEM of three experiments, **P < 0.01, ***P < 0.001. **(E)** Representative transmission electron microscopy images of negatively stained NC101 and NC101$^{\Delta NC-SV}$ cells 60 minutes post-infection with phage RSB03 (MOI 1). Scale bars represent 500 nm (50 nm in the inset). The final panel depicts an NC101$^{\Delta NC-SV}$ cell undergoing phage-induced lysis, with loss of cell envelope integrity.

data indicate that the NC-SV prophage does not have a fitness cost for *in vitro* growth in nutrient replete conditions, but does regulate a cell surface factor to reduce adsorption of some phage types to *E. coli* NC101.

## The NC-SV prophage is not required for intestinal colonization or extraintestinal dissemination

Although the NC-SV prophage did not confer a growth advantage in planktonic growth conditions *in vitro* (Fig 1C), we hypothesized that deletion of the NC-SV prophage would impact colonization fitness in the more stringent environment of the mammalian gut [40]. Studies of the well-characterized *E. coli*–phage λ system illustrate this point: frequent prophage reactivation and lysogeny exert variable effects on *E. coli* colonization fitness, outcomes that depend on the presence of susceptible bacteria [36].

First, we tested whether the NC-SV prophage conferred a fitness advantage for colonization of the intestine in mice with an intact microbiota. We colonized specific pathogen-free (SPF) C57BL/6J mice, which do not have baseline Entero-bacteriaceae colonization, with $10^8$ CFU of WT NC101 and NC101$^{\Delta NC\text{-}SV}$ [41]. Loss of NC-SV had no statistically significant effect on gut colonization kinetics (Fig E in S1 Text). Extraintestinal dissemination to the liver and spleen was rare for both strains (Fig E in S1 Text), and all mice appeared healthy appearing following colonization. Next, we compared colonization in mice with a defined microbiota [42] to assess whether the NC-SV prophage confers a fitness advantage in mice colo-nized with the highly similar *E. coli* AIEC strain Mt1B1, which has been previously shown to mediate colonization resis-tance to other Enterobacteriaceae driven by competition for competition for key carbon sources [43]. The Mt1B1 chromosome shares 99.99% average nucleotide identity with NC101 and encodes the same three NC101 prophages described herein (Fig F in S1 Text). In this setting, existing high-level Mt1B1 gut colonization conferred colonization resis-tance to subsequent NC101 colonization (Fig E in S1 Text), and the NC-SV deletion strain was more rapidly lost than the NC101 parental strain (Fig E in S1 Text), suggesting the NC-SV prophage confers a competitive fitness advantage in the intestine between highly similar strains.

Next, we evaluated the impact of NC-SV prophage loss on intestinal colonization and extraintestinal dissemination in monoassociated gnotobiotic mice. Germ-free mice have a lower barrier to extraintestinal dissemination of microbes due to relatively impaired mucosal barrier function [44–46], which is relevant to various disease states [47]. Germ-free 129SvEv mice were monoassociated with either NC101 or NC101$^{\Delta NC\text{-}SV}$. We observed no differences in body weight (Fig G in S1 Text), intestinal colonization (Fig G in S1 Text), or dissemination to extraintestinal sites (Fig G in S1 Text) between the two groups. Notably, both NC101 and NC101$^{\Delta NC\text{-}SV}$ disseminated to the liver and spleen in this setting, though no overt signs of invasive bacterial infection were detected during the study period. These findings indicate that the NC-SV prophage does not influence bacterial colonization or dissemination capacity in the simplified context of germ-free monoassociation.

## The NC-SV prophage restricts phage replication in the mouse intestine

Although prophage mediated inhibition of infection by competing phages is well characterized *in vitro*, infection dynamics in more complex ecological systems such as the gut are less well characterized. To test activity of RSB03 in the mouse intestine, we colonized specific pathogen-free C57BL/6 male mice with NC101 and administered RSB03 or heat-inactivated RSB03 enterally (Fig H in S1 Text). Fecal *E. coli* densities remained stable and comparable between groups (Fig H in S1 Text), indicating that acute RSB03 administration did not affect bacterial colonization density. RSB03 titers were quantifiable in fecal pellets from NC101-colonized mice that received viable RSB03, but remained undetectable in those that received heat-killed RSB03 (Fig H in S1 Text). In mice given active RSB03, PFUs were detected transiently after administration and again between days 9 and 12 (Fig H in S1 Text). These results are consistent with lytic replication of RSB03 in the intestinal environment.

Building on our *in vitro* findings, we hypothesized that the NC-SV prophage protects *E. coli* NC101 from phage infection *in vivo*. To test this, germ-free C57BL/6 mice were colonized with NC101 or NC101$^{\Delta NC\text{-}SV}$ and treated daily with RSB03 phage for five days (Fig I in S1 Text). Fecal *E. coli* density dropped approximately 2.6 $\log_{10}$ CFU/g during treatment but

remained stable thereafter in both groups, with no significant differences observed between strains at either day 4, day 15 (Fig 2A–2C), or at experimental endpoint (day 17) (Figs 2G and I in S1 Text).

In contrast, phage titers in fecal pellets were significantly higher in NC101$^{\Delta NC-SV}$ colonized mice during and after treatment, with RSB03 detectable in most NC101$^{\Delta NC-SV}$ colonized mice throughout the experiment, but only transiently or not at all in those colonized with NC101 (Figs 2D–2F and I in S1 Text). At endpoint (day 17), although there were no significant differences in *E. coli* density between strains in the gut (Figs 2G and I in S1 Text), NC101$^{\Delta NC-SV}$ mice exhibited higher phage loads throughout the gut including: the colon, small intestine, fecal contents, mucus, and tissue (Figs 2H and I in S1 Text). In contrast, NC101$^{\Delta NC-SV}$ mice exhibited significantly lower *E. coli* CFU in extraintestinal tissues such as the liver and spleen (Figs 2G and I in S1 Text) without significant differences in phage load (Figs 2H and I in S1 Text). Phage:host ratios were significantly elevated in mucus and tissue niches compared to feces (Figs 2I and I in S1 Text), suggesting these sites may increase phage replication or reduce bacterial survival. These data indicate that NC-SV limits phage replication in the gut and may indirectly influence bacterial dissemination to or survival in extraintestinal tissues by modulating phage pressure.

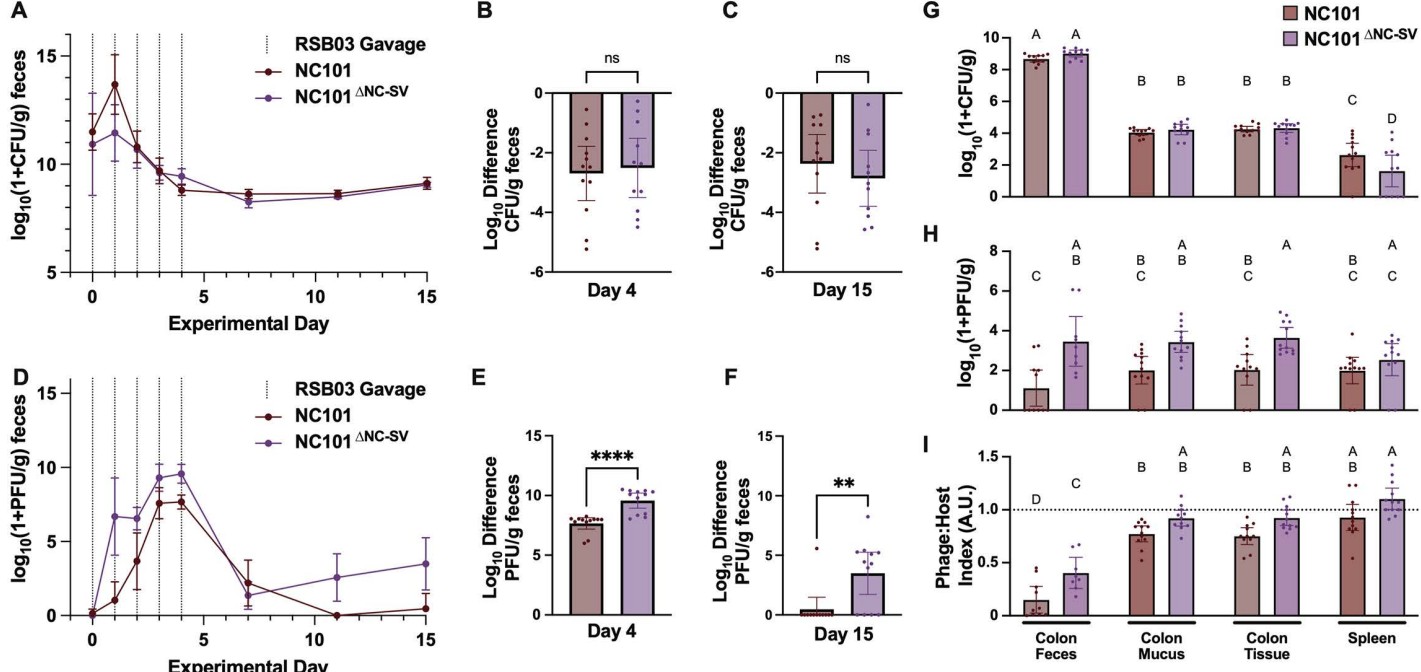

**Fig 2. The NC-SV prophage restricts RSB03 phage replication and persistence in the mouse gut.** Germ-free C57BL/6J mice were stably monoassociated with either NC101 or NC101$^{\Delta NC-SV}$ *E. coli*, then administered 1x10$^8$ PFU RSB03 daily by gavage for five consecutive days. **(A)** Serial quantification of *E. coli* log$_{10}$ CFU from fecal pellets plated on selective LB-chloramphenicol agar, normalized to fecal sample weight (g, mean +/- 95% confidence interval). Vertical dotted lines indicate days of RSB03 administration; P > 0.05, Mixed-effects model with Geisser-Greenhouse correction. **(B-C)** Log$_{10}$ difference in *E. coli* density in fecal pellets from baseline to **(B)** day 4 (last day of RSB03 treatment) and **(C)** day 15; ns, no statistically significant difference between groups. P > 0.05, unpaired nonparametric Mann-Whittney test. **(D)** Serial quantification of plaque forming units (log$_{10}$ PFU) from fecal pellets, normalized to fecal sample weight (grams, mean ± - 95% confidence interval); P < 0.0001, Mixed-effects model with Geisser-Greenhouse correction. **(E-F)** Log$_{10}$ difference in PFU in fecal pellets from baseline to **(E)** day 4 and **(F)** day 15; ****P < 0.0001, **P < 0.01, unpaired nonparametric Mann-Whitney test. **(G-I)** Detectable **(G)** *E. coli* log$_{10}$ CFU, **(H)** RSB03 log$_{10}$ PFU, and **(I)** Phage:Host index (min-max normalized log$_{10}$ difference of PFU-CFU, with 1 reflecting equal density) for the indicated samples at endpoint (day 17). For bar plots, dots represent individual subjects, bars represent the mean ± 95% confidence interval. Significantly different groups are indicated by compact letter display, where uppercase letters above each bar denote significantly different (P < 0.05) groups by two-way ANOVA and post hoc means testing (Tukey) comparing all groups. Groups that share a letter designation are not statistically different (e.g., A is significantly different from B, but not AB).

## The NC-SV prophage downregulates maltodextrin transport genes and reduces phage adsorption to *E. coli* NC101

To gain a deeper understanding of how the NC-SV prophage affects bacterial responses to phage infection, we performed RNAseq on wild-type NC101 and NC101$^{\Delta NC-SV}$ cells ten minutes post-infection with phage RSB03 (MOI = 1). Overall, 235 bacterial genes were significantly (P < 0.05) differentially regulated at least ±two-fold in NC101 compared to NC101$^{\Delta NC-SV}$ (Fig 3A and S1 Data). Gene enrichment analyses revealed that genes associated with carbohydrate transport, amino acid metabolism, and central carbon metabolism were significantly (P < 0.05) over-represented (Fig 3B).

Among these, maltodextrin transport genes (*lamB, malE, malF, malG, malK*; Fig 3A) were consistently downregulated in the presence of NC-SV. LamB and its associated transporter components are best known for their roles in maltodextrin uptake, and LamB is a receptor for several coliphages including λ, K10, TP1, Φ21, and Bp7 [21,22,48–50]. Even though RSB03 titers were unaffected by *lamB* deletion in planktonic growth conditions (Fig 3C), plaque areas were reduced in the NC101$^{\Delta NC-SV}$Δ*lamB* strain compared to NC101$^{\Delta NC-SV}$ (Fig 3D). While these findings indicate that LamB is not an essential receptor for RSB03, LamB appears to contribute to plaque expansion, perhaps through effects on carbohydrate transport and metabolism or outer membrane physiology.

We hypothesized that if NC-SV is regulating LamB expression to reduce phage adsorption to *E. coli* NC101, then lambda phage should exhibit infection phenotypes similar to those of phage RSB03 on *E. coli* NC101. However, lambda failed to infect *E. coli* NC101 or NC101$^{\Delta NC-SV}$. This outcome might be due to mutations in the NC101 LamB protein that are localized to its extracellular region [51]—the primary site of phage interaction [52] (Fig J in S1 Text). Alternatively, it is possible that the other prophages in the NC101 genome (NC-MV or NC-Ino) provide protection through superinfection exclusion or other mechanisms that suppress lambda replication, which could contribute to the observed lambda resistance in NC101 independent of LamB sequence differences.

Because carbohydrate availability is a key determinant of maltodextrin transporter expression, we next tested whether nutrient supplementation altered RSB03 infection dynamics. On lawns of NC101, glucose supplementation significantly reduced RSB03 plaque size compared to unsupplemented controls (Fig 4A), consistent with glucose-mediated repression of the maltodextrin transporter *lamB*. In contrast, maltodextrin supplementation enhanced plaque size relative to both unsupplemented and glucose-treated conditions (Fig 4A), indicating that induction of maltodextrin utilization pathways facilitates RSB03 spread. Glucose similarly reduced RSB03 adsorption to NC101 cells (Fig 4B), whereas maltodextrin modestly increased adsorption of both RSB01 and RSB03 but not RSB02 or RSB04 (Figs 4 and K in S1 Text). Interestingly, while RSB01 and RSB03 are related phages (Fig B in S1 Text), RSB01 adsorption was less affected by glucose supplementation than RSB03 (Figs 4C and K in S1 Text). This difference may reflect variation in how each phage engages LamB or the ability of RSB01 to utilize an alternative or secondary receptor under certain conditions. Taken together, these findings indicate that carbohydrate availability modulates NC101 phage susceptibility and that the maltodextrin transporter LamB contributes to, but is not solely responsible for, RSB03 adsorption and plaque growth.

## An NC-SV-encoded small RNA transcriptionally regulates maltodextrin transport and carbohydrate metabolism genes and reduces phage adsorption to *E. coli* NC101

Because NC-SV deletion altered competing phage adsorption, we hypothesized that NC-SV encodes gene(s) that influence phage adsorption to *E. coli* NC101. Prior work demonstrates that another lambdoid prophage called e14, which integrates into the same genomic locus as NC-SV (the *icd* gene), encodes a small RNA (sRNA) called co293 that alters cellular metabolism [53–58]. Specifically, co293 post-transcriptionally regulates the transcription factors HcaR and FadR [53], resulting in the upregulation of propionate degradation pathways and downregulation of glycolysis and the TCA cycle [53,59].

We hypothesized that NC-SV may encode a co293-like sRNA that modulates bacterial sugar transport and phage adsorption. To test this, we aligned the co293 sRNA sequence from *E. coli* MG1655 to the NC-SV genome and identified a

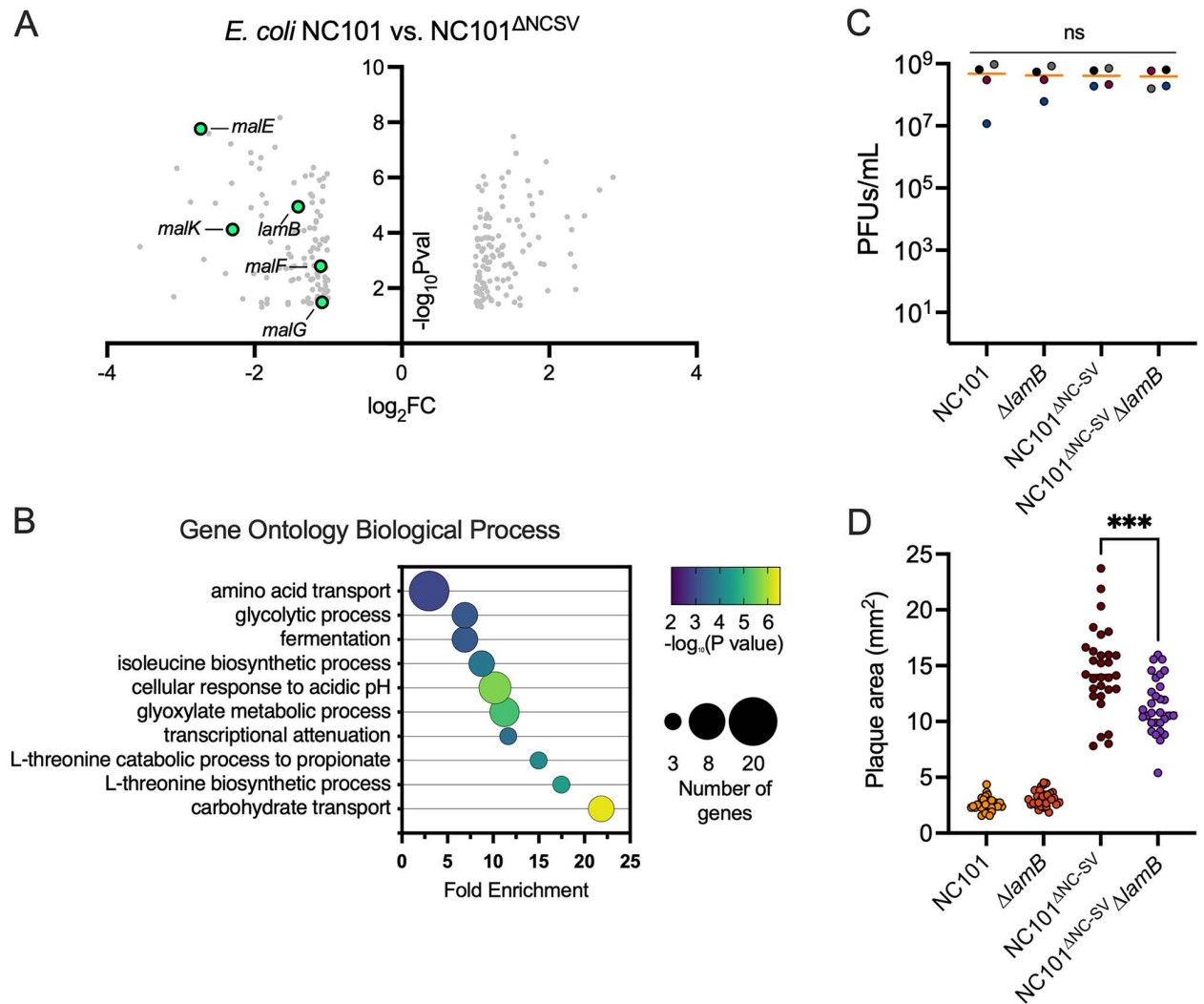

**Fig 3. The NC-SV prophage transcriptionally regulates maltodextrin transport and carbon metabolism in response to phage infection.** RNAseq was performed on mid-log NC101 or NC101^ΔNC-SV cells growing in LB broth 10 minutes post infection with phage RSB03 (MOI = 1). **(A)** Volcano plot showing the 235 differentially expressed genes in NC101 compared to NC101^ΔNC-SV with a ± 2-fold change and P ≤ 0.05. Data are representative of qua-druplicate experiments. **(B)** Gene enrichment analysis was performed on significantly differentially regulated genes shown in panel **A**. **(C)** Phage titers (PFU/mL) recovered from RSB03 infections of the indicated strains in broth culture. Data are the mean + /-SEM of three independent experiments. **(D)** Plaque area of RSB03 on lawns of NC101, ΔlamB, NC101^ΔNC-SV, or NC101^ΔNC-SVΔlamB. Plaques area was measured using Image J after 18 h of growth; each point represents one plaque, data are the mean + /-SEM of three independent experiments, ****P < 0.0001, ns = not significant.

77-nucleotide region with 58% sequence identity (45/77) located at the 3′ end of the *cI* phage repressor gene on the pos-itive strand (Fig 5A and 5B). The *cI* repressor is a conserved transcriptional regulator in lambdoid phages that maintains lysogeny by repressing lytic gene expression [60]. We refer to this candidate regulatory RNA as *svsR* (NC-SV sRNA). Secondary structure prediction of *svsR* revealed a stable hairpin structure (Fig 5C), a common feature of bacterial sRNAs that facilitates RNA stability and target binding [61,62].

To evaluate the conservation of *svsR* among related phages, we examined *E. coli* genomes containing NC-SV–like prophages with ≥30% genome coverage. *svsR* was frequently present and often located within the *cI* repressor gene, as indicated by triangle symbols in Fig 5D. To evaluate the distribution of NC-SV–like prophages, we queried

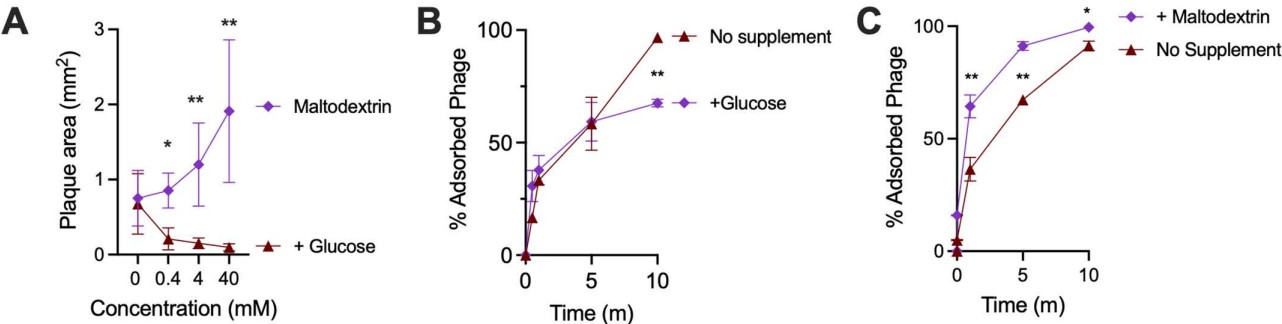

**Fig 4. Exogenous sugars modulate phage adsorption to *E. coli* NC101.** *E. coli* NC101 was grown in diluted (10%vol/vol) LB broth or agar supplemented with the indicated carbon sources at 37°C. **(A)** RSB03 plaque areas on NC101 lawns were measured after 18 hours. N = at least 30 plaques per condition. Statistical comparisons between glucose and maltodextrin at the same concentrations (0.4, 4, and 40 mM) were performed using unpaired Student's t tests; *P < 0.05, **P < 0.01. **(B and C)** The percentage of adsorbed RSB03 virions were measured in cultures supplemented with 40 mM of the indicated sugar at the indicated times post infection with an initial MOI of one. Data are the mean ± SEM of three experiments, *P < 0.05, **P < 0.01 by Student's t test.

the NC-SV genome against the NCBI *E. coli* RefSeq database. For our prophage prevalence analysis, we defined NC-SV–like prophages as those sharing ≥10% genome coverage with the NC-SV reference sequence. This threshold was chosen to balance sensitivity and specificity: it is stringent enough to exclude spurious short matches yet permissive enough to capture divergent prophages that still share conserved lambdoid gene modules. In practice, ≥ 10% coverage corresponded to tens of kilobases of aligned sequence—sufficient to encompass multiple conserved prophage structural and regulatory genes and reliably indicate the presence of related prophages. Using this cutoff, homologous sequences were detected in 4,114 *E. coli* strains. Based on BioSample metadata, isolates were classified by source as clinical (1,603), environmental (328), laboratory (109), or unknown (2,074) when no source information was available (Fig L in S1 Text).

Pathotype annotation revealed broad representation across multiple subgroups, including STEC/EHEC (n = 297), ETEC (n = 147), UPEC/ExPEC (n = 113), EPEC (n = 4), and laboratory K-12 lineage strains (n = 99), as well as AIEC (n = 2) (Fig L in S1 Text). The majority of strains (n = 3,452) lacked defined pathotype information and were classified as unassigned. These findings show that NC-SV–like prophages are not restricted to AIEC but instead occur widely across diverse *E. coli* lineages, including pathogenic, commensal, laboratory, and environmental isolates.

To determine whether NC-SV-related prophages are also prevalent beyond *E. coli*, we extended our analysis to a broader dataset of publicly available Enterobacteriaceae genomes. We identified NC-SV-like prophage sequences (≥10% genome coverage) in multiple genera, including *Klebsiella*, *Enterobacter*, *Citrobacter*, and *Salmonella* (Fig 5E). These findings suggest that NC-SV-like prophages—and *svsR*—are widespread across the Enterobacteriaceae family.

We next examined whether expression of the prophage-encoded sRNA svsR could reproduce the protective effects of the NC-SV prophage. In the absence of phage, *svsR* expression had little effect on exponential growth but produced a modest increase in final culture density compared to the empty-vector control (Fig 6A). Upon RSB03 infection, *svsR* reduced bacterial lysis relative to the control strain (Fig 6B). On lawns, *svsR*-expressing cells produced visibly smaller plaques (Fig 6C) with an average area roughly half that of the empty-vector control (Fig 6D). Consistent with reduced plaque sizes, svsR significantly decreased RSB03 adsorption to NC101$^{\Delta NC-SV}$ cells (Fig 6E). Notably, the plaque size phenotypes observed upon *svsR* expression in NC101$^{\Delta NC-SV}$ closely resemble those of WT NC101 reported in Fig 1: both exhibit smaller plaques and reduced phage adsorption compared to NC101$^{\Delta NC-SV}$. Collectively, these data demonstrate that *svsR* expression is sufficient to recapitulate the protective effects of the NC-SV prophage.

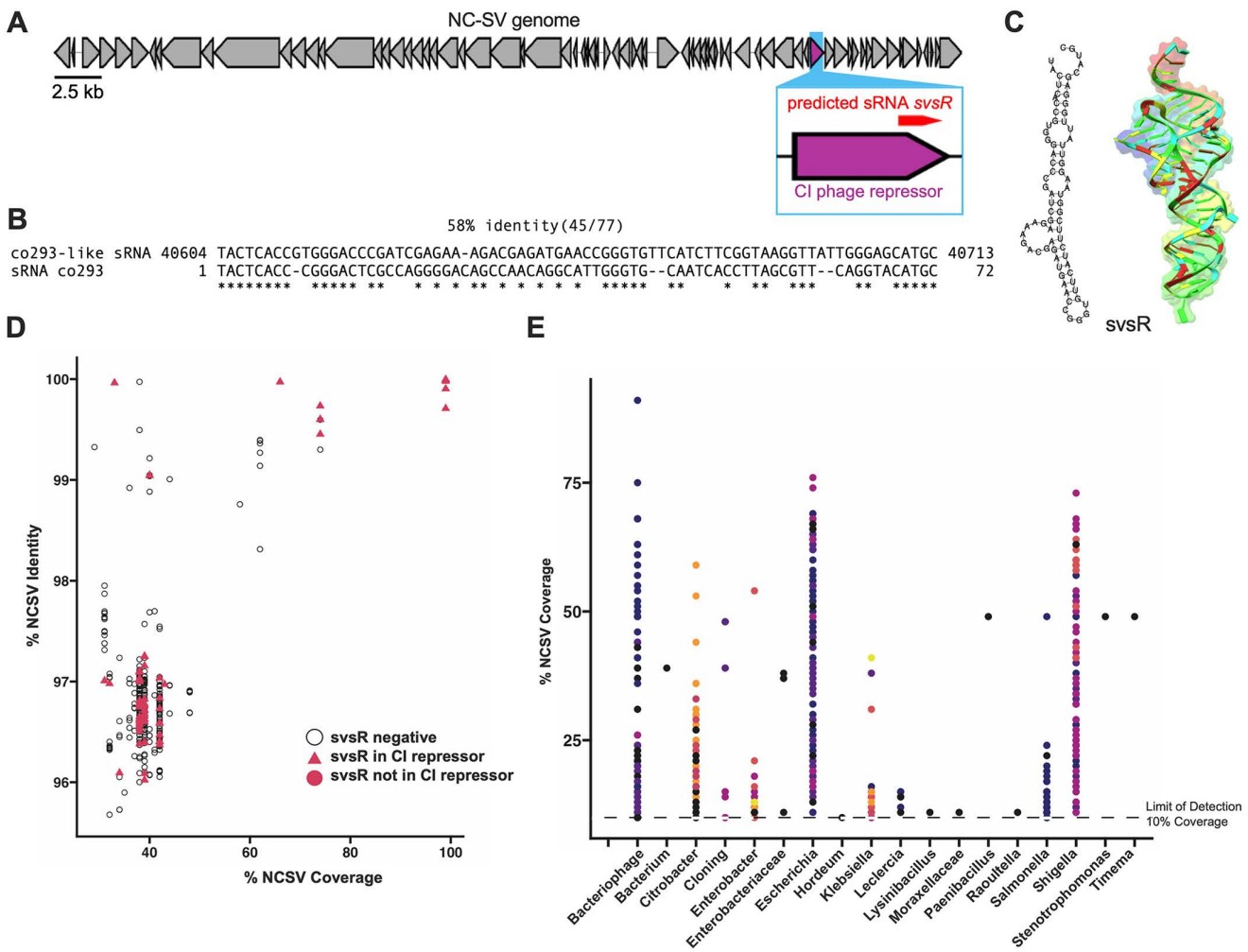

**Fig 5. The NC-SV prophage encodes a conserved co293-like small RNA, svsR, found across Enterobacteriaceae. (A)** Genome organization of the NC-SV prophage. The predicted *svsR* sRNA is encoded on the positive strand at the 3′ end of the *cI* phage repressor gene (highlighted in purple). **(B)** Sequence alignment between *svsR* and the *co293* sRNA from *E. coli* MG1655 reveals 58% identity (45/77 nucleotides). **(C)** Predicted secondary structure (left) and AlphaFold3 structural model (right) of *svsR*, reveal a stable hairpin conformation. **(D)** Conservation of *svsR* among NC-SV–like prophages in *E. coli* genomes with ≥30% NC-SV genome coverage. Each point represents a prophage, with % coverage plotted against % identity to the NC-SV reference. Open circles indicate the absence of *svsR*, while pink symbols indicate its presence; triangle shapes denote *svsR* embedded within the *cI* gene. **(E)** NC-SV–related prophages found in non-*E. coli* Enterobacteriaceae genomes with ≥10% genome coverage with each dot representing a genome.

Many sRNAs regulate gene expression by base-pairing with complementary sequences in target mRNAs [63]. However, a BLAST search of the NC101 genome revealed no complementary matches to *svsR*, suggesting that this sRNA may function through interactions with RNA-binding proteins rather than direct nucleic acid pairing.

To determine how *svsR* may affect the bacterial host independent of the NC-SV prophage, we performed RNAseq on NC101$^{\Delta NC-SV}$ cells carrying an empty expression vector or the *svsR* sRNA expression vector 10 minutes post infection by phage RSB03. In cells expressing *svsR*, 83 genes were significantly (P < 0.05) differentially regulated at least two-fold compared to the empty vector control (Fig 6F and S2 Data). Gene enrichment analyses revealed that genes associated with maltodextrin transport, glycerol metabolism, N-acetylglucosamine transport, and colanic acid biosynthesis (related to

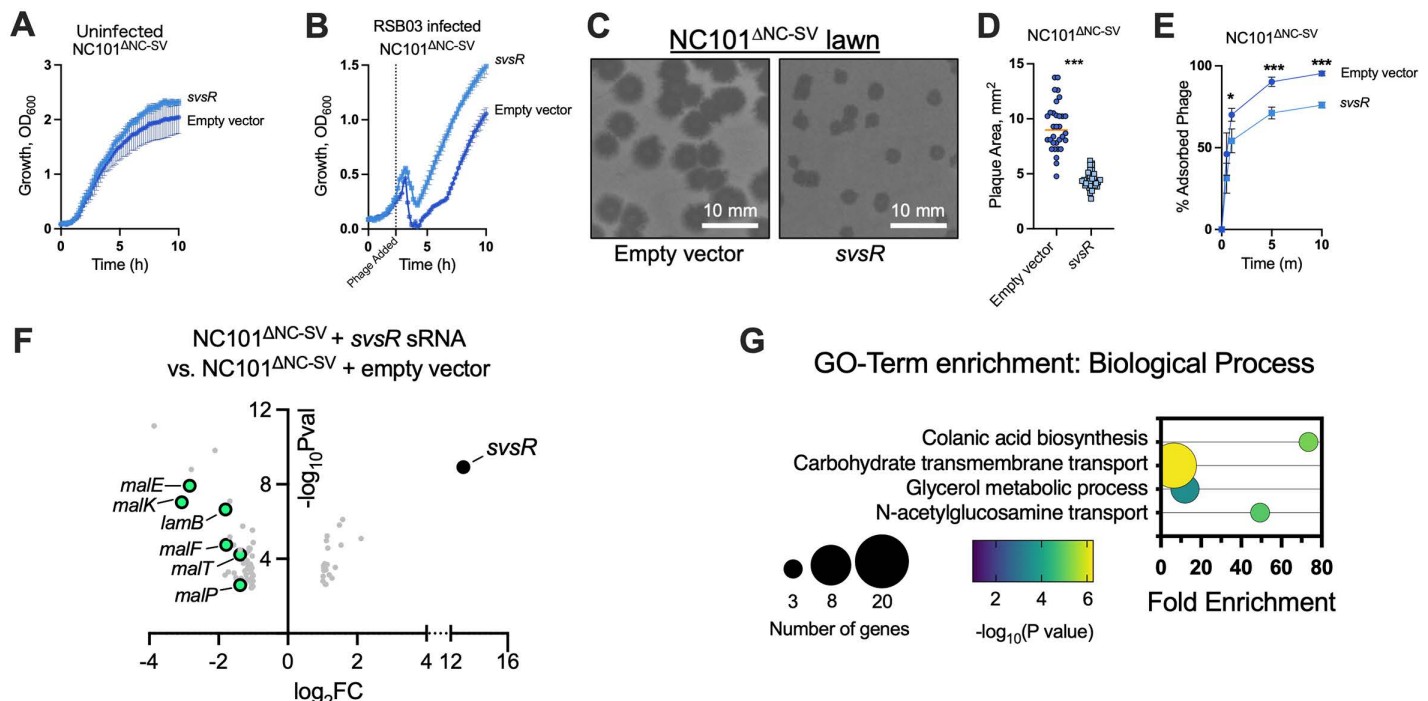

**Fig 6. The svsR sRNA downregulates maltodextrin transport genes and protects *E. coli* from phage infection. (A and B)** Growth (OD_{600}) of NC101^{ΔNC-SV} carrying an empty expression construct or a *svsR* sRNA expression construct was measured at the indicated times in (A) uninfected cells or (B) post infection with phage RSB03 at an MOI of one. Data are the mean ± SEM of three experiments. **(C)** Representative images of RSB03 plaques on the indicated lawns. **(D)** RSB03 plaque area was measured after 18 hours on lawns of NC101^{ΔNC-SV} expressing *svsR* in *trans* or empty vector control. **(E)** The percentage of RSB03 virions adsorbed to NC101^{ΔNC-SV} cells carrying the empty vector or the *svsR* expression vector was measured at the indicated times post infection with an initial MOI of one. Data are the mean ± SEM of three experiments; *$P < 0.05$,***$P < 0.001$. **(F)** Volcano plot showing the 83 differentially expressed genes in NC101^{ΔNC-SV} expressing *svsR* compared to the empty vector control with ±2-fold change and $P ≤ 0.05$. Data are representative of quadruplicate experiments. **(G)** Gene enrichment analysis was performed on the significantly differentially regulated genes shown in panel **F**.

biofilm formation in *E. coli* [64]) were significantly ($P < 0.05$) over-represented (Fig 6G). Specifically, maltodextrin transport genes (*lamB, malE, malF, malK, malP, malT*) were downregulated, similar to the pattern of maltodextrin transport genes downregulated by the NC-SV prophage (Fig 3). Additionally, other carbohydrate transport genes (*fruB, treB, treC, mglC, mglB, manX, manZ, manY, psuT, psuG*) were also downregulated (S2 Data) consistent with broader effects on bacterial carbohydrate metabolism.

Collectively, these results suggest that NC-SV, through the sRNA svsR, downregulates carbohydrate transport pathways, including *lamB* and maltodextrin uptake, which alters plaque growth and adsorption dynamics *in vitro*. In vivo, NC-SV is associated not only with reduced susceptibility of *E. coli* NC101 to RSB03, but also with altered bacterial dissemination in the presence of virulent phage pressure and a measurable fitness advantage for NC101 in the presence of competing *E. coli* strains.

## Discussion

Our study demonstrates that the lambda-like prophage NC-SV protects adherent-invasive *E. coli* from phage infection by downregulating maltodextrin and other carbohydrate transport genes. This regulation reduces virion adsorption to the bacterial surface and limits phage replication both *in vitro* and *in vivo*. These findings provide mechanistic insight into how prophages can modulate bacterial physiology to enhance survival under phage predation, which has implications for *E.*

*coli* persistence in phage-rich environments such as the inflamed IBD gut. Notably, NC-SV is also present in the prototypical AIEC strain LF82 [65], suggesting that its phage defense functions may be conserved among clinically relevant AIEC lineages relevant to IBD.

While prophage-conferred resistance to phages is a well-established outcome of lysogenic conversion [5], the diversity of mechanisms involved remains underexplored. Our work adds to this body of research by showing that NC-SV confers indirect resistance via transcriptional regulation of host carbohydrate transport systems. By repressing maltodextrin and other carbohydrate transport genes, NC-SV may alter the availability of carbohydrate-linked surface factors that can modulate phage adsorption.

Beyond phage resistance, NC-SV–mediated repression of maltodextrin transport may also modulate AIEC metabolic fitness. Dietary maltodextrins—prevalent in Western diets—enhance AIEC colonization [16] but also induce LamB expression, thereby increasing phage susceptibility [19,20]. Although our data show LamB is not required for RSB03 infection, the broader regulatory changes may still influence susceptibility to phage infection through other modifications to the bacterial cell surface. Thus, NC-SV may help AIEC navigate a trade-off between nutrient acquisition and phage vulnerability. This is consistent with emerging evidence linking bacterial metabolism to phage-host dynamics [66–69]. It is also possible that prophage-encoded superinfection immunity or related mechanisms suppress the replication of competing phages. Such prophage-mediated defenses are well described and could complement or act independently of the nutrient-based effects we observe.

Our *in vitro* findings led us to evaluate whether NC-SV also confers protection in more complex, host-associated environments. Importantly, intestinal *E. coli* density in SPF or germ-free mice colonized with NC101 was comparable to those colonized with NC101$^{\Delta NC-SV}$, indicating that NC-SV does not fundamentally affect bacterial fitness in the gut. The NC-SV prophage did confer improved colonization fitness in mice with baseline *E. coli* colonization, suggesting the NC-SV prophage plays a role in niche competition. Additionally, despite equivalent intestinal bacterial density between strains, RSB03 phage titers were significantly higher in feces and intestinal tissues of NC101$^{\Delta NC-SV}$ colonized mice, demonstrating that NC-SV restricts phage replication *in vivo*. Notably, elevated phage loads in NC101-colonized mouse intestines correlated with reduced *E. coli* burden in extraintestinal tissues such as the liver and spleen, suggesting that NC-SV-mediated control of phage replication may also enhance bacterial dissemination beyond the gut or persistence in extraintestinal sites in the presence of phage predation.

To identify molecular mechanisms underlying these effects, we examined NC-SV-regulated transcriptional responses during phage infection. Our RNA sequencing data revealed that an sRNA called *svsR* encoded by NC-SV downregulates maltodextrin transport genes. While the specific mechanism of action of this sRNA remains to be determined, sRNA-mediated gene regulation has emerged as an important mechanism for bacterial adaptation to environmental stressors. For example, in *E. coli* and *Salmonella enterica*, envelope stress induces the $\sigma^E$-dependent pathway, which regulates, among other genes, the expression of sRNAs that cause an abrupt downregulation in the expression of numerous outer membrane proteins [70]. In *S. enterica*, one of these outer membrane proteins is LamB, which is downregulated at the mRNA level by the sRNA *micA* in an Hfq-dependent manner [71]. Given that our BLAST analyses revealed no obvious base-pairing targets for svsR, it is possible that this sRNA functions through interactions with RNA-binding proteins such as Hfq or CsrA [72] rather than direct nucleic acid pairing.

The *svsR* sRNA is encoded on the 3' end of the *cI* gene. Notably, bacterial sRNAs are encoded within diverse genomic contexts, including the 3' ends of open reading frames. Some 3' sRNAs are transcribed independently from internal promoters, while others are processed from the 3' regions of mRNAs by RNase activity. For instance, the *sroC* sRNA in *E. coli* is derived from the 3' region of the *ygdR* mRNA and plays a regulatory role by acting as an RNA sponge for GcvB [73]. Similarly, *ryhB-2*, a homolog of the well-known iron-responsive sRNA *ryhB*, is transcribed from the 3' end of the *yhfA* coding region [74]. Additional work will be required to clarify whether svsR expression is controlled at the transcriptional or post-transcriptional level and to elucidate its biogenesis and functional mechanisms. Nonetheless, our findings

demonstrate that *svsR* plays a key role in remodeling carbohydrate transport and surface physiology, thereby conferring resistance to phage infection.

In addition to repressing *lamB* and other maltodextrin transport genes, svsR expression also altered transcription of genes involved in colanic acid biosynthesis (Fig 6G), a pathway known to impact bacterial survival in stressful environments. Colanic acid forms an exopolysaccharide capsule that can physically shield *E. coli* from phage predation by blocking receptor access [75]. Its production is also sensitive to carbon source availability, with glucose suppressing and other conditions enhancing its expression [76]. Moreover, colanic acid has been implicated in increased serum resistance and enhanced survival in host-associated environments [77]. Thus, svsR-mediated modulation of colanic acid biosynthesis may further improve *E. coli* resistance to phage infection and promote survival or dissemination beyond the gut, as observed in our *in vivo* experiments.

While its role in phage defense is clear, the specific targets of svsR remain unknown. For example, the sRNA encoded by the related e14 prophage targets the transcription factors *hcaR* and *fadR*, yet these genes were not differentially expressed in any of our RNAseq datasets, suggesting they are unlikely to be direct targets of svsR. Future studies will be needed to identify the regulatory targets and molecular interactions of svsR, as well as to determine whether it influences additional aspects of bacterial physiology beyond phage defense.

Overall, this work highlights how prophages can rewire bacterial metabolism to influence interactions with phages. Beyond classical phage defense systems (e.g., restriction modification, CRISPR-Cas, etc.), bacteria may exploit prophage-encoded regulatory elements to balance nutrient acquisition with phage resistance. These strategies likely support bacterial persistence in dynamic environments like the gut, where microbial communities are continually shaped by diet, interaction with the mammalian immune system, and phage predation. A deeper understanding of prophage-driven metabolic regulation could ultimately guide the development of dietary or pharmacological interventions to modulate microbial ecosystems and mitigate dysbiosis.

## Methods

### Bacterial and phage strains, growth conditions, plasmids, and primers

Bacterial strains, phages, plasmids, and their sources are listed in Table 2. Primer sequences are listed in Table 3. Unless indicated otherwise, bacteria were grown in LB at 37°C with shaking and supplemented with antibiotics or 0.1% arabinose

**Table 2. Bacterial strains, phages, and plasmids.**

| Strain | Source |
|---|---|
| NC101 | [24] |
| NC101<sup>ΔNC-Ino</sup> | This study |
| NC101<sup>ΔNC-MV</sup> | This study |
| NC101<sup>ΔNC-SV</sup> | This study |
| NC101:TN7-GFP | This study |
| NC101<sup>ΔNC-SV:TN7-GFP</sup> | This study |
| **Phages** | |
| RSB01 | This study |
| RSB02 | This study |
| RSB03 | This study |
| RSB04 | This study |
| **Plasmids** | |
| pSLTS | Addgene, [78] |
| pT2SK | Addgene, [78] |
| pHERD20T | [79] |
| pHERD20T-sRNA | This study |

**Table 3. Primers.**

| Name | Sequence 5'-3' |
| --- | --- |
| NCSVdel-For | TGATTGCGCCTTCCATACCT |
| NCSVdel-Rev | ATGGAAGGCGCTAAACTGCT |
| NCInodel-For | GCACGGTCGGTTAAGCCG |
| NCInodel-Rev | CACCACGAGCAGTTTGTTAG |
| NCMVdel-For | CACAACCGCAGAACTTTTCC |
| NCMVdel-Rev | GCTTGATGCGTTTATTGAGG |
| pHA-For | CGCAGGAAAGAACATGTG |
| pHA-Rev | AAGGGCCTCGTGATACG |
| M-For | ATCTCAAGAGTGGCAGC |
| M-Rev | TTACGCCCCGCCCTGC |
| pHA-Seq-For | TATCAGGGTTATTGTCTCATGAGCG |
| pHA-Seq-Rev | ACTTGAGCGTCGATTTTTGTGATGC |

when appropriate. Unless otherwise noted, antibiotics were used at the following concentrations: gentamicin (10 or 30 μg mL$^{-1}$), ampicillin (100 μg mL$^{-1}$), and carbenicillin (300 μg mL$^{-1}$).

## Wastewater phage isolation and sequencing

Samples were collected from the headwaters of the Missoula, Montana wastewater treatment plant. Collected samples were then plated onto lawns of *E. coli* NC101. Individual plaques were picked, plaque purified, and propagated in NC101 growing in LB broth. DNA was isolated from plaque purified phage isolates and sequenced by SeqCoast Genomics, LLC (Portsmouth, NH, USA). Phage genomes were assembled by SPAdes [80] and visualized using clinker [81].

## Construction of NC101 prophage mutants

Prophages NC-Ino, NC-MV, and NC-SV were individually deleted from the NC101 genome using pSLTS mediated genome editing [78]. Briefly, 5' and 3' ends of prophage mutation cassettes were designed with homology regions upstream and downstream of each prophage to be deleted and ordered as gene blocks (Azenta). Completed mutation cassettes for each prophage were constructed through Gibson Assembly using the primers listed in Table 3 to fuse 5' and 3' ends to a I-SceI cleavable kanamycin selectable marker and plasmid backbone from plasmid pT2SK. Completed prophage mutation cassettes were then introduced to NC101 cells containing the pSLTS helper plasmid via electroporation. pSLTS containing NC101 cells were induced with 2 mM L-arabinose to induce the expression of the lambda red recombinase to aid in recombination of mutation cassettes into the NC101 genome. Resulting colonies were screened for kanamycin resistance and sequenced to confirm successful mutation cassette integration. Once prophage deletions were confirmed, the kanamycin resistance marker was removed by plating PBS-resuspended colonies onto LB-amp plates containing 100 μg/mL anhydrotetracycline to induce the expression of the pSLTS encoded I-SceI endonuclease. Resulting colonies were then sequenced to confirm scarless deletions of each NC101 prophage.

## Plaque assays and plaque surface area measurements

Phage plaque assays were performed using a standard soft-agar overlay method. Briefly, the indicated bacterial strains were grown overnight in LB medium at 37 °C with shaking. The following day, cultures were diluted (1:100) into fresh LB and grown to mid-log phase (OD$_{600}$ ≈ 0.3–0.5). For each dilution of phage sample, 0.1 mL of phage suspension (serially diluted in phosphate-buffered saline) was mixed with 0.3 mL of host bacteria in soft agar (3 mL of molten soft agar, 0.6–0.7% agar, maintained at ~50 °C). After gently mixing, the soft agar suspension was overlaid onto prewarmed LB

agar plates (1.5% agar base). Plates were allowed to solidify at room temperature, then incubated inverted at 37 °C for 18 hours. Discrete plaques were visualized and imaged. Individual plaques were outlined using ImageJ and plaque surface areas (in mm²) were measured.

### Growth curves

Overnight cultures of the indicated strains were diluted to an $OD_{600}$ of 0.05 in 96-well plates containing LB and, if necessary, the appropriate antibiotics. After 3 h of growth at 37°C, strains were infected with the indicated phage and growth measurements resumed. $OD_{600}$ was measured using a CLARIOstar (BMG Labtech) plate reader at 37°C with shaking prior to each measurement.

### Phage adsorption assays

Phage adsorption assays were performed as previously described [82]. Briefly, an 18 mL test tube containing 5 mL of LB was inoculated with 100 µL of an overnight culture of the indicated strain. The culture was incubated at 37°C with shaking at 200 RPM for 1 hour until it reached a density of approximately $1 \times 10^8$ cells/mL (OD~0.2). Fresh lysate containing $1 \times 10^8$ phages ($N_{total}$) was then added, and the mixture was incubated at 37°C without shaking. At the indicated times, 500 µL aliquots were collected and centrifuged at 4,000g for 1 minute at room temperature to pellet adsorbed phages. Dilutions of the supernatant from the centrifuged sample (representing unabsorbed phage) were plated to determine phage titers. The total phage count ($N_{total}$) and the free phage count ($N_{free}$) were calculated from plate counts using the equation: $N_{adsorbed} = N_{total} - N_{free}$.

### Nutrient supplementation assays

LB broth was diluted with water to 10% vol/vol. For experiments where LB agar plates were used, the dilute LB was supplemented with agar (1.5% w/vol). Glucose or maltodextrin was then supplemented to the dilute LB broth or LB agar to a final concentration of 40 mM. Plaque assays or phage adsorption assays were then performed as described above.

### Mouse Husbandry

All mouse experiments were performed in compliance with federal regulations and guidelines set forth by the University of Utah Institutional Animal Care and Use Committee. All mice were housed under a 12-hour light/dark cycle per day. Germ-free (GF) C57BL/6 mice were originally obtained from the University of Michigan Mouse Facility and GF 129SvEv mice were originally obtained from the Gnotobiotic mouse facility at the University of North Caroline at Chapel Hill. Both were subsequently bred and maintained in the Gnotobiotic Core Facility at the University of Utah. GF and gnotobiotic PhyLo11B community mice (described in more detail below) are housed in sterile isolators with designated filtered air supply and screened bimonthly for contaminating microbes by both culture and 16S/18S PCR to detect bacterial and fungal organisms. Monoassociated GF mice and experimental PhyLo11B mice were removed from isolators to sterile Tecniplast cages with a designated filtered air supply for colonization experiments.

### Mouse model of intestinal adherent-invasive *E. coli* infection and enteral RSB03 treatment

**NC101 colonization of SPF mice.** Female adult specific pathogen free (SPF) C57BL/6J mice were ordered from Jackson Labs and housed in conventional cages with 5 mice per cage. Fecal samples from all mice were screened for detectable Enterobacteriaceae by plating on MacConkey Agar prior colonization of all mice in a cage with NC101 or NC101$^{\Delta NC-SV}$ ($1 \times 10^8$ CFU administered via daily oral gavage on experimental days 0–4). Fecal pellets were serially collected to quantify *E. coli* density. Mice were euthanized 2–3 weeks following initial colonization, and luminal intestinal contents and tissues were analyzed for *E. coli* CFU as detailed below. Results represent two independent experiments.

**NC101 colonization of PhyLo11B gnotobiotic mice.** We utilized PhyLo11B gnotobiotic C57BL/6J mice which harbor a defined Phylogenetically diverse, Low richness 11 Bacteria microbiota community which was engineered to replicate phylogenetic diversity normally present in mice while maintaining a low species richness [42]. PhyLo11B mice are maintained in a dedicated gnotobiotic isolator with stable community bacterial representation over successive generations [42]. Three independent cohorts of 8–14 week old PhyLo11B mice were inoculated with NC101 (n = 15 total; 8 females, 7 males) or NC101$^{\Delta NC-SV}$ (n = 15 mice total; 9 females, 6 males) via oral gavage of 1E8 CFU *E. coli* for four consecutive days (days 0–4). Fecal pellets were serially collected and plated on (i) MacConkey Agar to quantify total Enterobacteriaceae (reflecting PhyLo11B strain Mt1B1 and NC101) and (ii) LB chloramphenicol agar to quantify NC101 strains.

**NC101 monoassociation of germ-free mice.** 10-week old germ-free 129SvEv mice were transferred to sterile Tecnicages and inoculated with NC101 or NC101$^{\Delta NC-SV}$ (n = 8 for each strain, 4 females and 4 males) via single oral gavage (1x10$^8$ CFU, day 0). Mice were serially weighed along with collection of fecal pellets for CFU quantification of NC101 on selective LB-chloramphenicol agar. Mice were euthanized 16-weeks after NC101 colonization; intestines and tissues were collected for NC101 CFU quantification.

**RSB03 treatment of NC101 colonized SPF mice.** 6-week-old male specific pathogen free C57BL/6J mice (HK n = 5, active RSB03 n = 5) were ordered from Jackson Labs and housed in conventional cages with 5 mice per cage. 1x10$^8$ CFU *E. coli* were administered to each mouse via oral gavage for four consecutive days (days -29 to -26). Fecal pellets were collected once weekly following colonization to assess baseline *E. coli* density and confirm lack of baseline lytic phage activity in fecal pellets. On day 0, 3x10$^7$ PFU of RSB03 (or an equivalent volume of the heat-inactivated RSB03 from the same preparation) were administered by oral gavage in two doses, 7 hours apart. Fecal pellets were collected prior to the second dose of RSB03, then every 2–3 days for CFU and PFU quantification.

**RSB03 treatment of NC101 monoassociated gnotobiotic mice.** Two cohorts each containing 12 mice (6 female, 6 male) were housed with 2–4 mice per cage. Germ-free C57BL/6J mice were removed from germ-free isolators and placed into sterile Tecniplast cages at 6–8 weeks of age. Germ-free gnotobiotic mice were screened via 16S PCR to verify germ-free status prior to experimental use. On day -3, each mouse was gavaged with 1x10$^8$ *E. coli* (either NC101-GFP or NC101$^{\Delta NC-SV-GFP}$) as a single dose. Fecal pellets were collected three days following *E. coli* inoculation to quantify baseline *E. coli* density in feces and to confirm lack of baseline lytic phage activity in fecal pellets. From day 0 to day 4, RSB03 (1x10$^8$ PFU) was administered daily by oral gavage for five total doses. Fecal pellets were collected daily for *E. coli* CFU and PFU quantification prior to daily oral phage gavage. Additional fecal pellets were collected for analysis on experimental days 7, 11, and 15 following completion of RSB03 treatment. Intestinal contents and tissues, liver, and spleen were collected for CFU and PFU quantification on day 17.

## Quantification of CFU and PFU from mouse fecal pellets, mucus, and tissue samples

Fecal pellets were longitudinally collected from live mice. Intestinal luminal contents from the colon and small intestine were removed following euthanasia for analysis at endpoints. Fecal samples were homogenized in HBSS with Ca$^{2+}$/Mg$^{2+}$ (Corning 21–023-CV) by pipet disaggregation and vortexing for 30 seconds, centrifuged at 50 x g for 10 minutes to pellet fecal debris, and the supernatant was removed (fecal wash). Fecal wash samples were centrifuged at 2,500 x g for 10 minutes to pellet bacteria. The fecal wash supernatant (bacteriophage fraction) was centrifuged at 7,500 x g, 10 minutes, 4°C to pellet any residual bacteria, and supernatant was serially diluted in HBSS with Ca$^{2+}$/Mg$^{2+}$ for PFU quantification. The remaining bacterial pellet was washed with HBSS, then resuspended in HBSS with Ca$^{2+}$/Mg$^{2+}$ (bacterial fraction) and serially diluted for CFU enumeration.

Intestinal tissues were thoroughly rinsed three times in PBS. To digest intestinal mucus, 1000 μl of HBSS without Ca$^{2+}$/Mg$^{2+}$ (Corning 21–022-CV) with 10 mM HEPES and 1.5 mM DTT was added to each sample and incubated with shaking for 15 min at room temperature (mucus fraction). Intestinal tissue fragments were removed from mucus digestion and blotted to remove excess liquid. All tissues were homogenized in bead-beating tubes containing two sterile 6.35 mm stainless

steel beads (BioSpec) and 0.3% Triton-X/ HBSS with $Ca^{2+}/Mg^{2+}$ using the Omni Bead Ruptor Bead Mill Homogenizer for 1 min, 4 m/s at room temperature. For PFU quantification, the mucus fraction and tissue homogenate were spun at 7500 x g, 10 min, 4ºC to pellet bacteria and debris and supernatant was serially diluted for PFU quantification (phage fraction).

For CFU quantification: The mucus fraction and tissue homogenates were serially diluted 1:10 in HBSS with $Ca^{2+}/Mg^{2+}$ and plated on LB-chloramphenicol (30 μg/mL) plates. For fecal samples, 10 μl of each dilution was plated at four 1:10 serial dilutions, and the dilution resulting in countable single colonies was used to determine CFU/g feces.

For PFU quantification: For all phage fractions, 100 μl of the phage fraction and 100 μl NC101 *E. coli* ($OD_{600}$ 0.5-0.7) was added to 5 mL 0.6% soft LB top agar and poured over LB agar plates. Phage fractions were diluted in HBSS with $Ca^{2+}/Mg^{2+}$ as needed to obtain isolated plaques for counting. Plates were incubated overnight (12–18 hours) prior to counting phage plaques.

### RNA purification and RNA-seq

Total RNA was extracted from the indicated strains and conditions 10 minutes post phage infection using TRIzol. The integrity of the total cellular RNA was evaluated using RNA tape of Agilent TapeStation 2200 before library preparation. All RNA samples were of high integrity with a RIN score of 7.0 or more. rRNA was first depleted from 500 ng of each sample using MICROBExpress Kit (AM1905, Fisher) following the manufacture's instruction. The rRNA-depleted total RNA was subjected to library preparation using NEBNext Ultra II RNA Library Prep Kit (E7700, New England Biolabs) and barcoded with NEBNext Multiplex Oligos for Illumina (E7730, New England Biolabs) following the manufacturer's instructions. The libraries were pooled with equal amounts of moles, further sequenced using MiSeq Reagent V3 (MS-102–3003, Illumina) for pair-ended, 600-bp reads, and demultiplexed using the build-in bcl2fastq code in Illumina sequence analysis pipeline. Raw sequencing reads have been deposited as part of BioProject PRJNA1225266 in the NCBI SRA database.

### RNA-seq data analysis

RNAseq data analysis were performed as previously described [83]. Briefly, RNA-seq reads were aligned to the reference *E. coli* NC101 genome (GenBank: GCA_029542525.1), mapped to genomic features, and counted using Rsubread package v1.28.1 [84]. Count tables produced with Rsubread were normalized and tested for differential expression using edgeR v3.34.1 [85]. Genes with ≥twofold expression change and a *P* value below 0.05 were considered significantly differential. Functional classification and Gene Ontology (GO) enrichment analysis were performed using PANTHER classification system (http://www.pantherdb.org/) [86]. RNA-seq analysis results were plotted with GraphPad Prism version 10.

### Structure prediction

AlphaFold3 [87] was used to predict the structures of LamB trimers from *E. coli* NC101 and MG1655 or the putative sRNA encoded by NC-SV. Structures were visualized with ChimeraX-1.9 [88,89].

### Transmission electron microscopy imaging

Cells were grown to mid-log phase ($OD_{600} \approx 0.4$), infected with RSB03 at a multiplicity of infection of 1 for 1 h, harvested, and washed with PBS. Samples were fixed in 4% paraformaldehyde, applied directly to carbon-coated copper grids, and negatively stained with 2% (wt/vol) uranyl acetate. No embedding or thin sectioning was performed; intact cells were visualized on the grid surface. Grids were imaged on a Tecnai Spirit 120 kV transmission electron microscope.

### Identification and classification of NC-SV prophage homologs

The complete NC-SV genome was used as a BLASTn query against all *Escherichia coli* assemblies in the NCBI Assembly database. Assemblies containing at least one region with ≥ 90% nucleotide identity and ≥ 70% query coverage were

classified as NC-SV-positive. For each positive assembly, the accession number, contig, coordinates, and alignment statistics were recorded. Assembly accessions were linked to their associated BioSample records using the NCBI E-utilities API. Metadata fields (e.g., strain, host, isolation source, disease, collection date, and geographic origin) were parsed from XML records. Each isolate was assigned a pathotype based on keyword matching of BioSample attributes and strain names. Terms corresponding to STEC/EHEC, ETEC, UPEC/ExPEC, EPEC, AIEC, and laboratory lineages (e.g., "K-12", "MG1655", "BW25113", "DH5α") were used to classify strains; others were labeled Unassigned. Isolates were further grouped by source category (Clinical, Environmental, Laboratory, or Unknown) based on metadata describing the isolation source.

## Statistical analyses

Unless specified otherwise, differences between datasets were evaluated by Student's t test, using GraphPad Prism version 10.6.1. Longitudinal statistical comparisons between groups from *in vivo* experiments were performed using a mixed effects model with Geisser-Greenhouse correction, with individual timepoints assessed by Sidak's multiple comparisons test. For *in vivo* experiments, a nonparametric Mann-Whitney test to compare ranks was used for comparisons between two groups, and two-way ANOVA with Tukey's multiple comparisons test was used for statistical comparisons between multiple groups. For all tests, a P value $< 0.05$ was considered statistically significant.

## Supporting information

**S1 Text. Fig A. Comparative analysis of NC-SV and lambda prophages.** Genomic alignment between the NC-SV prophage and phage Lambda. Arrows represent predicted open reading frames. Homologous open reading frames are connected by ribbons shaded in gray-scale according to pairwise amino-acid identity (0–100%). **Fig B. RSB wastewater phage genome comparisons.** Phages RSB01–04 were isolated from the headwaters of the wastewater treatment plant in Missoula, Montana, USA. Phage RSB01 (44,523 bp; NCBI accession PX402465) and RSB03 (43,192 bp; NCBI accession PX402467) are both Veterinaerplatzviruses that share 51.7% nucleotide homology, while phages RSB02 (76,472 bp; NCBI accession PX402466) and RSB04 (76,787 bp; NCBI accession PX402468) are related Kuraviruses that share 60.7% nucleotide homology. The genome comparisons shown here were generated by Clinker and homologous open reading frames are connected by ribbons shaded in gray-scale according to pairwise amino-acid identity (0–100%). **Fig C. Phage titers of RSB01–RSB04 on NC101 and prophage-deletion mutants.** *E. coli* NC101 and its isogenic prophage deletion derivatives (NC101$^{\Delta NC-SV}$, NC101$^{\Delta NC-MV}$, and NC101$^{\Delta NC-Ino}$) were infected with wastewater-derived phages RSB01–RSB04. Phage titers were quantified as plaque-forming units per milliliter (PFU/mL) 6 h post-infection. Data are the mean±SEM from three independent experiments, *P$< 0.05$, determined by unpaired Student's t test comparing PFUs on each mutant strain to wild-type NC101; ns, not significant. **Fig D. The NC-SV prophage protects *E. coli* NC101 from infection by some, but not all phages by reducing virion adsorption. (A)** Representative images of plaques formed by wastewater phage isolates RSB01, RSB02, or RSB04 on WT NC101 or the indicated prophage mutants are shown. **(B-D)** The surface area of plaques formed on lawns of the indicated strains was measured, N=50 plaques per condition, from four replicate experiments, ****P$< 0.0001$; ns=not significant. **(E-G)** Growth of WT NC101 and NC101$^{\Delta NC-SV}$ was measured after infection with the indicated phages at an MOI of one. **(H-J)** The percentage of adsorbed virions was measured in WT NC101 or NC101$^{\Delta NC-SV}$ cells at the indicated times post infection, MOI=1. Data are the mean±SEM of three experiments, ***P$< 0.001$. **Fig E. The NC-SV prophage is dispensable for intestinal colonization of mice without baseline *E. coli* colonization. (A–B)** Specific pathogen-free (SPF) C57BL/6J mice were screened to confirm lack of baseline Enterobacteriaceae from fecal pellets, then stably colonized with the indicated NC101 *E. coli* strains via daily oral gavage ($10^8$ CFU, days 0–4 indicated by vertical dashed lines, n=10 female mice/group). **(A)** Quantification of NC101 colony forming units ($\log_{10}$ CFU/g feces, mean±95% CI) from fecal pellets plated on selective LB-chloramphenicol agar. P=0.2725, mixed-effects model with Geisser-Greenhouse correction. **(B)** Detectable $\log_{10}$ CFU mean±- 95% CI for the

indicated specimens collected after 2–3 weeks of colonization. Significantly different groups are indicated by corresponding letter design, where uppercase letters above each bar denote significantly different (P<0.05) groups by two-way ANOVA and post hoc means testing (Tukey) comparing all groups. Groups that share a letter designation are not statistically different (e.g., A is significantly different from B, but not AB). **(C–D)** The NC-SV prophage confers intestinal colonization fitness in the setting of baseline *E. coli* colonization. C57BL/6J PhyLo11B gnotobiotic mice stably colonized with a defined consortia of 11 bacteria (including the *E. coli* strain Mt1B1) were gavaged with the indicated NC101 strains daily via oral gavage (10^8 CFU, days 0–4 indicated by vertical dashed lines, n=15 mice/group, mixed male and female mice). (C) NC-SV deletion impairs competitive intestinal colonization. Quantification of NC101 colony forming units (log10 CFU/g feces, mean±95% CI, yellow and orange lines) from fecal pellets plated on selective LB-chloramphenicol agar compared to total *E. coli* CFU quantified on MacConkey Agar (representing Mt1B1 and NC101, magenta and pink lines). ****P<0.0001, ns, not significant; mixed-effects model with Geisser-Greenhouse correction. (D) Loss of detectable NC101 is hastened in the strain lacking the NC-SV prophage in Phy01t18 gnotobiotic mice with high baseline *E. coli* colonization. The percentage of mice with consistent detectable NC101 from selective plating of feces on LB-chloramphenicol agar is plotted by for each strain. P<0.01, Mantel-Cox proportional hazard test. **Fig F. Comparative genome maps of E. coli NC101 and Mt1B1.** Chromosomes of NC101 and Mt1B1 were annotated using Prokka. The two strains share 99.99% average nucleotide identity, and both encode the three NC101 prophages described in this study. Gene organization and prophage positions are highly conserved between the strains, underscoring their close genetic relatedness. **Fig G. The NC-SV prophage is dispensable for intestinal monocolonization of gnotobiotic mice.** The NC-SV prophage does not affect intestinal colonization or extraintestinal dissemination in gnotobiotic mice monocolonized with NC101. 129Sv/Ev germ-free (GF) mice were colonized with NC101 (n=8) or NC101$^{\Delta NC-SV}$ (n=8) *E. coli* via single oral gavage at 10-weeks of age (10$^8$ CFU, day 0). **(A)** Weight gain normalized to starting weight. P=0.13, mixed-effects model with Geisser-Greenhouse correction. **(B)** Quantification of NC101 colony forming units (log10 CFU/g feces, mean+/- 95% CI) from fecal pellets plated on selective LB-chloramphenicol agar. P=0.54, mixed-effects model with Geisser-Greenhouse correction. **(C)** Detectable log10 CFU (mean+/- 95% CI) for the indicated specimens collected 4 months following initial colonization. Significantly different groups are indicated by compact letter display, where uppercase letters above each bar denote significantly different (P<0.05) groups by two-way ANOVA and post hoc means testing (Tukey) comparing all groups. Groups that share a letter designation are not statistically different (e.g., A is significantly different from B, but not AB). SI, small intestine. CFU, colony forming units. **Fig H. Phage RSB03 replicates in the mouse gut. (A)** Schematic overview of the experimental design created with BioRender.com. Specific pathogen-free (SPF) C57BL/6J mice were stably colonized with NC101 *E. coli* followed by oral administration of either active or heat-inactivated RSB03 phage (3x10$^7$ PFU per dose, 2 doses, 7-hours apart). Created in BioRender. Bell, R. (2026) https://BioRender.com/jg98yai. **(B)** Quantification of *E. coli* colony forming units (log$_{10}$ CFU/g feces, mean+/- 95% CI) from fecal pellets plated on selective LB-chloramphenicol agar; there were no statistically significant differences between strains. **(C)** Quantification of phage plaque-forming units (log$_{10}$ PFU/g feces, mean+/- 95% CI) from fecal pellets; mice that received active RSB03 had significantly higher detectable PFU (P<0.0001, mixed-effects model with Geisser-Greenhouse correction) which was transiently detected the day of RSB03 administration and up to 12 days after administration. **Fig I. The NC-SV prophage is associated with reduced E. coli dissemination to extraintestinal tissues and lower phage loads in the gut of monoassociated gnotobiotic mice. (A)** Schematic overview of experimental design. Germ-free C57BL/6J mice were stably colonized with either NC101 or NC101$^{\Delta NC-SV}$ *E. coli*, then administered 1x10$^8$ PFU RSB03 daily by gavage for five consecutive days. Fecal pellets, luminal contents, and tissues were collected for quantification of colony-forming units (CFU) and plaque forming units (PFU). Created in BioRender. Bell, R. (2026) https://BioRender.com/r8m130m. **(B-D)** Detectable *E. coli* CFU (**B**), RSB03 PFU (**C**) and Phage:Host index (**D**, min-max normalized log difference of PFU-CFU, with 1 reflecting equal density) for the indicated samples at endpoint. SI, small intestine. For bar plots, dots represent individual subjects, bars represent the mean±95% confidence interval. Significantly different groups are indicated by compact letter display, where uppercase letters above each bar denote significantly different (P<0.05) groups by two-way ANOVA and post hoc means testing (Tukey) comparing all groups. Groups that share a letter designation are not

statistically different (e.g., A is significantly different from B, but not AB). **Fig J. Comparison of LamB sequences and structures from *E. coli* NC101 and *E. coli* K-12 (1MAL). (A)** LamB protein sequences were aligned using Clustal. Amino-acid differences between the two sequences are highlighted in red. The N-terminal signal peptide and periplasmic residues (1–25) shown in gray were omitted from the solved LamB crystal structure (PDB 1MAL). **(B)** Predicted LamB trimers from *E. coli* NC101 (AlphaFold model, orange) and the experimentally determined LamB structure (PDB 1MAL, green) are displayed as surface renderings. Residues that differ between NC101 and 1MAL LamB are shown in hot pink. **Fig K. The impact of glucose and maltodextrin on phage virion adsorption to *E. coli* NC101.** The percentage of adsorbed RSB01, RSB02, and RSB04 virions were measured at the indicated times post infection with an initial MOI of one. Data are the mean±SEM of three experiments. **Fig L. Distribution of *E. coli* genomes carrying NC-SV–like prophages by source and pathotype. (A)** Source categories of *E. coli* assemblies containing NC-SV–like prophages, classified from NCBI BioSample metadata. Clinical isolates were the largest defined group (n=1,603), followed by environmental (n=328) and laboratory reference strains (n=109). A substantial fraction (n=2,137) lacked sufficient metadata for source assignment (*Unknown*). **(B)** Pathotypes of *E. coli* genomes with NC-SV–like prophages identified by BLASTn (≥ 10% genome coverage) across 4,114 *E. coli* assemblies. Most strains were unassigned (n=3,515), but NC-SV–like prophages were also detected among multiple pathogenic subgroups, including STEC/EHEC (n=297), ETEC (n=147), UPEC/ExPEC (n=113), EPEC (n=4), as well as laboratory K-12 lineage strains (n=99) and the AIEC model strain NC101 (n=2).
(DOCX)

**S1 Data. Differential gene expression in NC101 vs. NC101ΔNC-SV during early phage infection.** This dataset contains RNAseq results comparing *Escherichia coli* NC101 (wild type) and NC101$^{\Delta NC-SV}$ during early infection (10 minutes) with phage RSB03 at a multiplicity of infection (MOI) of 1. RNA was collected from mid-log phase cultures 10 minutes post-infection. The table includes normalized gene expression values (counts per million, CPM), $\log_2$-fold changes, exact test P-values, and false discovery rate (FDR)–adjusted P-values generated using the edgeR package. Genes with ≥2-fold differential expression and $P < 0.05$ were considered significantly changed.
(XLSX)

**S2 Data. Transcriptomic analysis of svsR expression in NC101ΔNC-SV during phage infection.** This dataset contains RNAseq results comparing NC101$^{\Delta NC-SV}$ cells carrying an empty vector or the svsR expression construct during early infection (10 minutes) with phage RSB03 (MOI=1). RNA was collected 10 minutes post-infection from mid-log phase cultures. The table provides normalized gene expression values (CPM), $\log_2$-fold changes, exact test P-values, and FDR-adjusted P-values generated using the edgeR package. Genes with ≥2-fold differential expression and $P < 0.05$ were considered significantly changed.
(XLSX)

## Author contributions

**Conceptualization:** Patrick R. Secor.

**Formal analysis:** Nicole L. Pershing, Robert S. Brzozowski, Blake Wiedenheft, Sherwood R. Casjens, June L. Round, Patrick R. Secor.

**Funding acquisition:** June L. Round, Patrick R. Secor.

**Investigation:** Nicole L. Pershing, Robert S. Brzozowski, Amelia K. Schmidt, Dominick R. Faith, Alex C. Joyce, Lizett Ortiz de Ora, John D. Kominsky, Annika Dankwardt, Andrew Maciver, Rickesha Bell, William S. Henriques, Shelby E. Andersen, Breck A. Duerkop, June L. Round.

**Methodology:** Nicole L. Pershing, Robert S. Brzozowski, Amelia K. Schmidt, Dominick R. Faith, Shelby E. Andersen.

**Project administration:** June L. Round, Patrick R. Secor.

**Supervision:** Blake Wiedenheft, Breck A. Duerkop, June L. Round, Patrick R. Secor.

**Writing – original draft:** Nicole L. Pershing, June L. Round, Patrick R. Secor.

**Writing – review & editing:** Nicole L. Pershing, Blake Wiedenheft, Sherwood R. Casjens, Breck A. Duerkop, June L. Round, Patrick R. Secor.

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
