## [Decision Letter · Decision Letter 0]

22 Jun 2025

A prophage-encoded sRNA limits lytic phage infection of adherent-invasive E. coli

PLOS Pathogens

Dear Patrick,

Thank you for submitting your manuscript to PLOS Pathogens. After careful consideration, we feel that it is an interesting topic and has considerable merit but does not fully meet PLOS Pathogens's publication criteria as it currently stands. Therefore, we invite you to submit a revised version of the manuscript that addresses the points raised during the review process.

Please submit your revised manuscript within 60 days Aug 21 2025 11:59PM. If you will need more time than this to complete your revisions, please reply to this message or contact the journal office at plospathogens@plos.org. Please include the following items when submitting your revised manuscript:

We look forward to receiving your revised manuscript.

John

John M Leong

Academic Editor

PLOS Pathogens

David Skurnik

Section Editor

PLOS Pathogens

Sumita Bhaduri-McIntosh

Editor-in-Chief

PLOS Pathogens

orcid.org/0000-0003-2946-9497

Michael Malim

PLOS Pathogens

orcid.org/0000-0002-7699-2064

**Journal Requirements:**

At this stage, the following Authors/Authors require contributions: Robert S. Brzozowski, Amelia K. Schmidt, Nicole L. Pershing, Annika Dankwardt, Dominick R. Faith, Alex C. Joyce, Andrew Maciver, William S. Henriques, Shelby E. Andersen, Blake Wiedenheft, Sherwood R. Casjens, Breck A. Duerkop, June L. Round, and Patrick Secor. Please ensure that the full contributions of each author are acknowledged in the "Add/Edit/Remove Authors" section of our submission form.

https://journals.plos.org/plospathogens/s/submission-guidelines#loc-parts-of-a-submission

4) We do not publish any copyright or trademark symbols that usually accompany proprietary names, eg ©,  ®, or TM  (e.g. next to drug or reagent names). Therefore please remove all instances of trademark/copyright symbols throughout the text, including:

- ® on page: 20

- TM on page: 20.

5) We notice that your supplementary Figures are included in the manuscript file. Please remove them and upload them with the file type 'Supporting Information'. Please ensure that each Supporting Information file has a legend listed in the manuscript after the references list.

Potential Copyright Issues:

i) Figures S4A, and S5A. Please confirm whether you drew the images / clip-art within the figure panels by hand. If you did not draw the images, please provide (a) a link to the source of the images or icons and their license / terms of use; or (b) written permission from the copyright holder to publish the images or icons under our CC BY 4.0 license. Alternatively, you may replace the images with open source alternatives. See these open source resources you may use to replace images / clip-art:

7) Thank you for stating 'Raw sequencing reads have been deposited as part of BioProject PRJNA1225266 in the NCBI SRA database.' Please note that, though access restrictions are acceptable now, your entire minimal dataset will need to be made freely accessible if your manuscript is accepted for publication. This policy applies to all data except where public deposition would breach compliance with the protocol approved by your research ethics board. If you are unable to adhere to our open data policy, please kindly revise your statement to explain your reasoning and we will seek the editor's input on an exemption.

8) Please amend your detailed Financial Disclosure statement. This is published with the article. It must therefore be completed in full sentences and contain the exact wording you wish to be published.

**Reviewers' Comments:**

Reviewer's Responses to Questions

**Part I - Summary**

Reviewer #1: They find that a lambdoid prophage confers protection against virulent phages. This is an important finding and worthy of PloS Pathogens. The manuscript is very well written. However, more data are needed to support their findings. Also, they lack a mechanism for how the sRNA is reducing virulent phage receptor expression. Papers that have sRNAs as the central player, should report on what their relevant target(s) are, in my opinion.

Reviewer #2: The manuscript by Brzozowski et al reports the discovery and characterization of a very interesting interaction between E. coli prophage and lytic phage through modulation of metabolism. The manuscript employs a satisfying mix of in vitro and in vivo (mouse) experiments, as well as RNA sequencing. In general, I like this study and found the manuscript to be well written and on a topic that is broadly relevant. I have some relatively minor comments listed below, the most important of which relates to the apparent use of parametric statistical tests in contexts (mouse data) where non parametric tests are likely more appropriate.

The authors should ensure that the statistical test used is stated in each figure legend (throughout). There is a stats section in the methods but specific information should additionally be in each figure legend. The compact letter display for statistical differences is not clear which comparisons are made and which are significant (e.g. Fig 2). In some cases (CFU and PFU etc data from mouse experiments like Fig 2 and S5 etc) the data do not look normally distributed, yet the methods state that the statistical tests used were t tests (parametric). It is strongly recommended to employ non parametric tests especially when the sample size is small (as here) and the assumptions of normal distribution cannot be formally assessed as they could properly be if sample sizes were very large.

Do the C57Bl/6 mice used (Fig S4A-B) have native E. coli in their microbiota? It seems from the methods that the CFU quantification involves selection on chloramphenicol agar plates for NC101(line 604 in methods) but I recommend specifying in the figure legend both for thoroughness and because if selective plating was not performed then native E. coli could have affected counts by adding to the E. coli load present. Regardless, it seems interesting to know if E. coli are present in the mouse gut prior to NC101 colonization since it could provide interesting ecological context for the study. Along these lines, profiling of the microbiota (e.g. 16S amplicon sequencing) could provide information about if the e coli colonization and/or phage treatment altered microbiome composition.

Figure 2 legend should indicate that mice are gnotobiotic (ex-germfree).

There is ~6 log higher E coli load in the gnotobiotic (ex germfree) mice (fig 2) compared with the SPF mice (Fig S4), with corresponding higher phage burden. Are the observed findings about extraintestinal dissemination and phage loads relevant when only they are only observed in a non-physiological context of E coli mono-colonization of gnotobiotic mice?

Do the gnotobiotic mice experience weight loss over the experiment? Dissemination to extraintestinal tissues seems likely to be accompanied by inflammation and this seems relevant.

For Clinker analysis, the coloring indicates common cluster membership of genes based on global sequence alignments, not functional modules as stated in Fig S1 legend.

The authors should include clearly stated NCBI accession numbers in Methods or in a supplementary table.

For plaque assays (e.g. Fig 1, S3), the authors should make clear if biological replicates were performed.

I am quite skeptical of the statement and references cited of glucose/maltodextrin concentrations in the intestinal lumen (line 289). 40mM seems quite high – most glucose is absorbed in the small intestine and I would expect glucose concentration in the colon lumen to be vanishingly low. Furthermore, dietary changes would likely alter the concentration significantly so I am wary that this 40mM value is accurate. Instead, I suggest that the author perform plaque experiments using a varied range of concentrations (perhaps 10 fold) to assess the question of glucose/maltodextrin levels on LamB expression and plaque size.

Figure S6, there is a PDB crystal structure of LamB: 1MAL. I recommend the authors utilize this rather than the AF predicted structure and cite the paper PMID 7824948.

There is no statement in the methods indicating that testing was performed to confirm germfree status prior to gavage of germfree mice with E. coli. I assume this was done but recommend stating for completeness.

Reviewer #3: In this manuscript, Brzozowski and colleagues focus on prophage infection of adherent-invasive E. coli (AIEC), NC101. They identify a lambda-like prophage, NC-SV, that confer protection against two lytic phages isolated from the headwaters of the Missoula, Montana wastewater treatment plant (!), RSB01 and RSB03, in vitro. This was confirmed for RSB03 in vivo in mice. Comparison of the transcriptomes of E. coli NC101 and NC101∆NC-SV 10 min post-infection identified over 200 genes that are differentially regulated. Genes associated with carbohydrate transport, amino acid metabolism and central carbon metabolism were over-represented. Based on LamB, a maltodextrin transporter, being the receptor for numerous coliphages the authors hypothesized that NC-SV is regulating LamB expression on the outer membrane to reduce phage adsorption. The study also identified a NC-SV encoded small RNA that regulates more than 80 E. coli genes, including maltodextrin transport genes. Overall, the manuscript is nicely written, easy to understand and with a logical flow. The figures nicely represent the experimental content. I also feel that the bulk of the conclusions were supported by the data. Overall, it is a solid and impactful study. I provide some suggestions below on how to bolster the robustness of the LamB-phage infection angle of the manuscript and points in the text that require clarification.

**Part II – Major Issues: Key Experiments Required for Acceptance**

Reviewer #1: 1. Because the NC-SV prophage could encode virulence or transmission factors, it is premature to conclude that sensitivity to predation by RSB03 is why the NC-SV deletion strain has a different bacterial load and dissemination in mice compared to the parent strain NC101. You must compare loads and dissemination for these two strains in mice in the absence of RSB03 phage.

2. The paper seems incomplete without knowing how the NC-SV-encoded sRNA downregulates LamB. Did you computationally search for putative targets of the sRNA?

3. Although there are differences in plaque area, there is no in vitro data regarding phage EOP? Wouldn’t you expect a change in EOP due to change in surface receptor expression?

Reviewer #2: (No Response)

Reviewer #3: (1) The maltodextrin and glucose addition experiments support that the phage receptor is regulated by carbohydrates. I suspect that many genes encoding carbohydrate transporters are regulated by these sugars, however. If MalB is the actual receptor for RSB103 (and RSB101), then an E. coli NC101∆malB strain should be resistant to infection by these lytic phages. This experiment is key to confirming that these phages bind to MalB on the bacterial surface.

(2) The altered production of MalB (i.e. at the protein level) should be confirmed for the ∆NC-SV strain, also for the ∆NC-SV sRNA strain compared to empty vector. Whole cell lysates would be great, surface expression would be even better.

**Part III – Minor Issues: Editorial and Data Presentation Modifications**

Reviewer #1: 1. Line 100: Where is the key for the color-coded functional modules? Same for Fig S2.

2. Line 122: Better to call them virulent phages throughout rather than lytic phages. Temperate phages have a lytic cycle, after all.

3. Figure 1:

A. Is �NC-MV plaque size also a bit smaller?

B. Panel D: move legend to side as it appears to be data points.

C. Max adsorption is reached by 10 min, so how much does this

difference matter at 5 min?

D. Panel E: It is unclear what is being shown here. Is the final panel showing a cell with a burst region of the cell envelope? The figure legend should say something to help the reader.

E. The conclusion that the cell surface receptor makes them more prone to lysis is one possibility. Couldn’t it also be that NC-SV encodes for a protein that triggers phage defense systems that recognize RSB01/03? If the cell surface receptor plays a role in membrane integrity, shouldn’t you show this through other methods such permeability to ethidium bromide or the like?

4. Figure 2:

A. The quality of the figure is bad. Too pixelated.

B. Panels G-I: there is no description in the legend to indicated what A, B, C or D mean above each bar graph (same comment for figure 5S).

5. Figure S6: Should you consider that inability of lambda phage to infect NC101 might also be due to phage super immunity that’s not related to the phage receptor, but suppression of phage cycle?

6. Figure S7: Why isn’t RSB01 adsorption being impacted by glucose supplementation? Does it have an alternative receptor?

7. Figure 6A: there’s a pretty big difference growthwise in empty vector vs svsR expressing cells. Why is this?

8. Figure 6: Although visually it appears svsR insertion behaves pretty much as WT from prior figures, it would be beneficial to compare it to WT NC101 to show if expression of svsR is sufficient to restore WT plaque area reduction and decrease in phage adsorption.

9. Material and methods: For the plaque assay, you don’t say how many cells were used or if they are in top agar, etc. Instead there is information regarding the mouse experiment.

Reviewer #2: (No Response)

Reviewer #3: (3) Lines 148-152. “Phage RSB01 and RSB03 adsorption was significantly (P<0.01) higher in NC101ΔNC-SV compared to NC101 cells (Fig 1D, Fig S3H) while adsorption of phages RSB02 and RSB04 was not affected (Fig S3I and J), indicating that RSB02 and RSB04 use a different cell surface receptor than RSB01 and RSB03.” It is surprising then that glucose reduces RSB03 adsorption to NC101 bacteria but minimal impact on RSB01. Please comment in the text as to why.

(4) It is not clear how prevalent the lambdoid prophage NC-SV is in E. coli strains. Is it restricted to adherent-invasive E. coli, for example? Please provide more background information about this aspect.

(5) Line 89. “Neatly missing”. Odd choice of words. Suggest changing.

(6) Table 1. It is difficult, if not impossible, to see the shading that indicates percent identity. Suggest changing graphics.

(7) A few y-axes have units listed as log(1+CFU/g), see Figure S4B and C, Figure 2G and H, Figure S5B and C for example. I believe this should be log10CFU/g.

(8) Figure S5 legend. I am confused about “Significantly different groups are indicated by compact letter display”. I see there are small letters above the columns but I don’t know what they mean.

(9) Line 262. Suggest changing sentence to “This outcome might be due to seven amino acid mutations in the C-terminus of MalB that are localized to the extracellular region – the primary site of phage interaction”. The alignment is much more striking than the existing verbiage.

(10) Line 342. Why was ≥10% genome coverage chosen? Please clarify in the text what this means?

(11) Line 621. Typo. Should be manufacturer’s.

(12) Transmission electron microscopy. I think there are some details missing. Were the bacteria embedded and sectioned?

PLOS authors have the option to publish the peer review history of their article (what does this mean? ). If published, this will include your full peer review and any attached files.

**Do you want your identity to be public for this peer review?** For information about this choice, including consent withdrawal, please see our Privacy Policy .

Reviewer #1: No

Reviewer #2: No

Reviewer #3: No

**Figure resubmission:**

**Reproducibility:**



---

## [Decision Letter · Decision Letter 1]

19 Dec 2025

Dear Pat,

We are pleased to inform you that your manuscript 'A prophage-encoded sRNA limits phage infection of adherent-invasive E. coli' has been provisionally accepted for publication in PLOS Pathogens.

Best regards,

John

John M Leong

Academic Editor

PLOS Pathogens

David Skurnik

Section Editor

PLOS Pathogens

Sumita Bhaduri-McIntosh

Editor-in-Chief

PLOS Pathogens

orcid.org/0000-0003-2946-9497

Michael Malim

Editor-in-Chief

PLOS Pathogens

orcid.org/0000-0002-7699-2064

Reviewer Comments (if any, and for reference):

Reviewer's Responses to Questions

**Part I - Summary**

Reviewer #2: The authors have responded well to my initial review comments and those of the other reviewers. I am satisfied with this revision and have no further comments.

Reviewer #3: The authors have addressed my previous concerns in the revised version of the manuscript.

**Part II – Major Issues: Key Experiments Required for Acceptance**

Reviewer #2: (No Response)

Reviewer #3: (No Response)

**Part III – Minor Issues: Editorial and Data Presentation Modifications**

Reviewer #2: (No Response)

Reviewer #3: (No Response)

PLOS authors have the option to publish the peer review history of their article (what does this mean? ). If published, this will include your full peer review and any attached files.

**Do you want your identity to be public for this peer review?** For information about this choice, including consent withdrawal, please see our Privacy Policy .

Reviewer #2: **Yes: ** Benjamin Ross

Reviewer #3: No

---

## [Editor Report · Acceptance letter]

Dear Dr. Secor,

We are delighted to inform you that your manuscript, "A prophage-encoded sRNA limits phage infection of adherent-invasive E. coli," has been formally accepted for publication in PLOS Pathogens.

Best regards,

Sumita Bhaduri-McIntosh

Editor-in-Chief

PLOS Pathogens

orcid.org/0000-0003-2946-9497

Michael Malim

Editor-in-Chief

PLOS Pathogens

orcid.org/0000-0002-7699-2064